# High-resolution structures of multiple 5-HT$_{3A}$R-setron complexes reveal a novel mechanism of competitive inhibition

Sandip Basak[1,2], Arvind Kumar[1,2], Steven Ramsey[3], Eric Gibbs[1,2], Abhijeet Kapoor[3], Marta Filizola[3], Sudha Chakrapani[1,2,4]*

[1]Department of Physiology and Biophysics, Case Western Reserve University, Cleveland, United States; [2]Cleveland Center for Membrane and Structural Biology, Case Western Reserve University, Cleveland, United States; [3]Department of Pharmacological Sciences, Icahn School of Medicine at Mount Sinai, New York, United States; [4]Department of Neuroscience, School of Medicine, Case Western Reserve University, Cleveland, United States

**Abstract** Serotonin receptors (5-HT$_{3A}$R) play a crucial role in regulating gut movement, and are the principal target of setrons, a class of high-affinity competitive antagonists, used in the management of nausea and vomiting associated with radiation and chemotherapies. Structural insights into setron-binding poses and their inhibitory mechanisms are just beginning to emerge. Here, we present high-resolution cryo-EM structures of full-length 5-HT$_{3A}$R in complex with palonosetron, ondansetron, and alosetron. Molecular dynamic simulations of these structures embedded in a fully-hydrated lipid environment assessed the stability of ligand-binding poses and drug-target interactions over time. Together with simulation results of apo- and serotonin-bound 5-HT$_{3A}$R, the study reveals a distinct interaction fingerprint between the various setrons and binding-pocket residues that may underlie their diverse affinities. In addition, varying degrees of conformational change in the setron-5-HT$_{3A}$R structures, throughout the channel and particularly along the channel activation pathway, suggests a novel mechanism of competitive inhibition.

*For correspondence:
Sudha.chakrapani@case.edu

Competing interests: The authors declare that no competing interests exist.

## Introduction

Cancer treatment by radiation or chemotherapy triggers the release of excess serotonin from the mucosal enterochromaffin cells in the upper gastrointestinal tract (*Schwörer et al., 1991*). Serotonin binds to serotonin (3) receptors (5-HT$_3$Rs), a pentameric ligand-gated ion channel (pLGIC), on the vagal afferent nerve in the gut and on the chemoreceptor trigger zone in the brainstem leading to severe nausea and vomiting in patients receiving cancer treatments. These common side effects of cancer treatments take a significant physical and psychological toll on cancer patients. Without management, these side effects can reduce patient compliance, undermining treatment success (*Gilmore et al., 2018*). Furthermore, uncontrolled debilitating side effects result in secondary complications such as dehydration and anorexia that require additional hospitalization and increase overall healthcare costs.

Current antiemetic therapies include a 5-HT$_3$R antagonist treatment regimen, which is considered a major advancement in improving patient quality of life during cancer treatment. Setrons, competitive antagonists of 5-HT$_3$R, are effective in the prevention of chemotherapy-induced nausea and vomiting (CINV), radiation therapy- induced nausea and vomiting (RINV), and postoperative nausea and vomiting (PONV) (*Spiller, 2011*; *Hsu, 2010*). Notably, CINV occurs in acute and delayed phases. The first generation of FDA approved setrons are effective for treating acute but not delayed phase nausea due to their short plasma half-lives. They belong to the following major classes based on

**eLife digest** Serotonin is perhaps best known as a chemical messenger in the brain, where it regulates mood, appetite and sleep. But as a hormone, serotonin works in other parts of the body too. Serotonin is predominantly made in the gut, where it binds receptor proteins that help to regulate the movement of substances through the gastrointestinal tract, aiding digestion. However, a surge in serotonin release in the gut induces vomiting and nausea, which commonly happens as a side effect of treating cancer with radiotherapy and chemotherapy.

Anti-nausea drugs used to manage and prevent the severe nausea and vomiting experienced by cancer patients are therefore designed to target serotonin receptors in the gut. These drugs, called setrons, work by binding to serotonin receptors before serotonin does, essentially neutralising the effect of any surplus serotonin. Although they generally target serotonin receptors in the same way, some setrons are more efficient than others and can provide longer lasting relief. Clarifying exactly how each drug interacts with its target receptor might help to explain their differential effects.

Basak et al. used a technique called cryo-electron microscopy to examine the interactions between three common anti-nausea drugs (palonosetron, ondansetron and alosetron) and one type of serotonin receptor, 5-HT3AR. The experiments showed that each drug changed the shape of 5-HT3AR, thereby inhibiting its activity to varying degrees. Further analysis identified a distinct 'interaction fingerprint' for the three setron drugs studied, showing which of the receptors' subunits each drug binds to. Simulations of their interactions also showed that water molecules play a crucial role in the process, exposing the binding pocket on the receptor's surface where the drugs attach.

This work provides a structural blueprint of the interactions between anti-nausea drugs and serotonin receptors. The structures could guide the development of new and improved therapies to treat nausea and vomiting brought on by cancer treatments.

their chemical structures: carbazole (*e.g.* ondansetron), indazole (*e.g.* granisetron), indole (e.g. dolasetron, tropisetron), and pyridoindole (*e.g.* alosetron). Although setrons share the same fundamental mechanism of action, they have varying efficacies, dose-response profiles, duration of action, and off-target responses. These differences perhaps underlie variable patient response, particularly in the context of acute and refractory emesis (*de Wit et al., 2005*). The isoquinoline derivative palonosetron, the only FDA approved second generation setron, is shown to have a longer half-life, improved bioavailability, and efficacy. In addition, palonosetron is implicated in causing receptor internalization, which further improves antiemetic properties. Beyond their role in controlling emesis, setrons are used to treat GI disorders including irritable bowel syndrome (IBS), obesity, and several inflammatory, neurological and psychiatric disorders such as migraine, drug abuse, schizophrenia, depression, anxiety, and cognitive disorders. However, in some cases, toxicity and adverse side effects have hampered their use. For example, the FDA approved treatment of diarrhea-predominant IBS with alosetron led to severe ischemic colitis in many patients (*Friedel et al., 2001*). Given the broad therapeutic potential of 5-HT$_{3A}$R antagonists, the prospect of substantial therapeutic gains by probing the setron pharmacophore as well as developing novel pharmaceuticals with higher efficacy and reduced side effects is encouraging.

At the physiological level, the 5-HT$_3$Rs play an important role in gut motility, visceral sensation, and secretion (*Engel et al., 2013*; *Lummis, 2012*; *Kia et al., 1995*; *Bétry et al., 2011*; *Thompson and Lummis, 2006*; *Gershon, 2004*), and are also implicated in pain perception, mood, and appetite. 5-HT$_3$Rs are the only ion channels (*Maricq et al., 1991*) among the large family of serotonin receptors, the rest being G-protein coupled receptors (GPCRs). 5-HT$_3$Rs are expressed as homopentamers of subunit A or heteropentamers of subunit A, in combination with B, C, D, or E subunits (*Niesler et al., 2007*). Compositional and stoichiometric differences lead to differential responses to serotonin, gating kinetics, permeability, and pharmacology (*Davies et al., 1999*; *Kelley et al., 2003*; *Thompson and Lummis, 2013*). This functional diversity, tissue specific expression patterns, and distinct pathophysiology of 5-HT$_3$R isoforms establish a need for subtype specific drugs to address diverse clinical needs (*Hammer et al., 2012*). Of note, granisetron, palonosetron, ondansetron, and alosetron have slightly different affinities for various receptor subtypes (*Gregory and Ettinger, 1998*). Ondansetron, in addition to binding to 5-HT$_3$Rs, also binds to several

GPCRs, such as 5-HT$_{1B}$R, 5-HT$_{1C}$R, α1-adrenergic receptors and μ-opioid receptors (*Kovac, 2016*). Granisetron binds to all subtypes of 5-HT$_3$R, but has little or no affinity for 5-HT$_1$R, 5-HT$_2$R and 5-HT$_4$R receptors. Palonosetron is highly selective for 5-HT$_{3A}$R and dolasetron for 5-HT$_{3AB}$R (*Smith et al., 2012*). Structural insights into setron-binding poses came initially from crystal structures of the acetylcholine binding protein (AChBP), bound to granisetron, tropisetron, or palonosetron (*Kesters et al., 2013*; *Hibbs et al., 2009*; *Price et al., 2016*) and more recently from 5-HT$_{3A}$R complexed with tropisetron and granisetron (*Polovinkin et al., 2018*; *Basak et al., 2019*). While some of the basic principles of setron-binding are now clear, there is still limited understanding of differing pharmacodynamics among setrons and the associated clinical relevance.

In the present study, we have solved cryo-EM structures of the full-length 5-HT$_{3A}$R in complex with palonosetron, ondansetron, and alosetron at the resolution range of 2.9 Å to 3.3 Å. Together with our previously solved structures of 5-HT$_{3A}$R in complex with granisetron (5-HT$_{3A}$R-Grani) and serotonin-bound (5-HT$_{3A}$R-serotonin, State 1-preopen), as well as ligand-free 5-HT$_{3A}$R (5-HT$_{3A}$R-Apo), we provide details of various setron-binding modes and the ensuing conformational changes that lead to channel inhibition. Using molecular dynamics (MD) simulations and electrophysiology, we have further validated setron-binding modes and interactions within the conserved binding pocket. Combined with abundant functional, biochemical, and clinical data, these new findings may serve as a structural blueprint of drug-receptor interactions that can guide new drug development.

## Results and discussion

### Cryo-EM structures of setron-5-HT$_{3A}$R complexes

Structures of the full-length 5-HT$_{3A}$R in complex with setrons were solved by single-particle cryo-EM. Detergent solubilized 5-HT$_{3A}$R was incubated with 100 μM of palonosetron, ondansetron, or alosetron for 1 hr prior to vitrification on cryo-EM grids. Iterative classifications and C5 symmetry-imposed refinement produced a final three-dimensional reconstruction at a nominal resolution of 3.3 Å for 5-HT$_{3A}$R-Palono (with 91,163 particles), 3.0 Å for 5-HT$_{3A}$R-Ondan (67,333 particles), and 2.9 Å for 5-HT$_{3A}$R-Alo (42,065 particles) (*Figure 1—figure supplement 1a and b*). The local resolution of the map was estimated using ResMap and in the range of 2.5–3.5 Å for each of these reconstructions (*Figure 1—figure supplement 1c*). Structural models were built using refined maps containing density for the entire extracellular domain (ECD), transmembrane domain (TMD), and the structured regions of the intracellular domain (ICD) (*Figure 1a* and *Figure 1—figure supplement 2*). Overall, each of the setron-bound 5-HT$_{3A}$R complexes has an architecture similar to previously solved 5-HT$_{3A}$ receptors (*Polovinkin et al., 2018*; *Hassaine et al., 2014*; *Basak et al., 2018a*; *Basak et al., 2018b*). Three sets of peripheral protrusions corresponding to N-linked glycans are bound to the Asn82, Asn148, and Asn164 in each subunit (*Figure 1a*, right). A strong, unambiguous density is seen at each of the intersubunit interfaces, corresponding to individual setrons (*Figure 1b*). Besides this site, no additional densities for setrons were found under these conditions, although there have been predictions that palonosetron may act as both an orthosteric and allosteric ligand (*Del Cadia et al., 2013*).

### Ligand-Receptor interactions

The map quality was particularly good at the ligand-binding site allowing us to model sidechains and the setron orientation. Setrons bind within the canonical neurotransmitter-binding pocket and are lined by residues from Loops A, B, and C on the principal (+) subunit and Loops D, E, and F from the complementary (-) subunit (*Figure 2* and *Figure 2—figure supplement 1*). Residues within 4 Å of setron include Asn101 in Loop A, Trp156 in Loop B, Phe199, and Tyr207 in Loop C, Trp63 and Arg65 in Loop D, and Tyr126 in Loop E. These residues are strictly conserved, and perturbations at each of these positions impact efficacy of setrons and serotonin (*Yan et al., 1999*; *Duffy et al., 2012*; *Thompson et al., 2005*). In each setron-5-HT$_{3A}$R complex, the essential pharmacophore of the setron is placed in a similar orientation: the basic amine is at the deep-end of the pocket in the principal subunit; the defining aromatic moiety interacts with residues in the complementary subunit; and the carbonyl-based linker, between the two groups, is essentially coplanar with the aromatic ring.

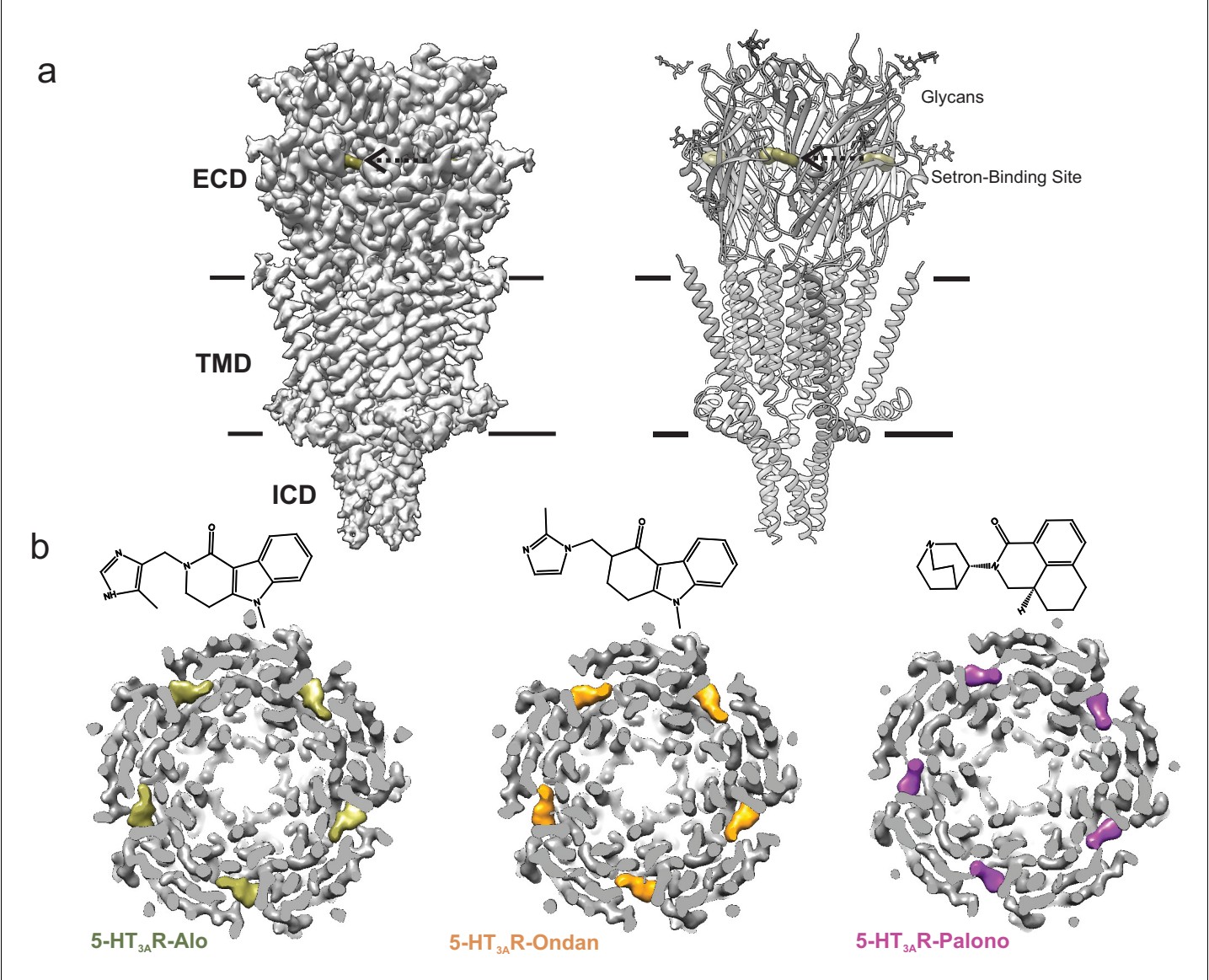

**Figure 1.** Cryo-EM structure of 5-HT$_{3A}$R-setron complexes. (a) Three-dimensional reconstruction of 5-HT$_{3A}$R-Alo at 2.92 Å resolution (*left*) and the corresponding structural model (*right*) that shows the overall architecture consisting of the extracellular domain (ECD), transmembrane domain (TMD), and structural regions of the intracellular domain (ICD). The alosetron density is shown in *deep olive* color and the three sets of glycans are shown as stick representation. Arrow points toward the setron density. Solid line denotes putative membrane limits. (b) Extracellular view of 5-HT$_{3A}$R-Alo (*left*), 5-HT$_{3A}$R-Ondan (*middle*), and 5-HT$_{3A}$R-Palono (*right*) maps sliced at the neurotransmitter-binding site. In each case, the five molecules of respective setrons are highlighted in colors. Chemical structures of setrons are shown above.

The online version of this article includes the following figure supplement(s) for figure 1:

**Figure supplement 1.** Resolution estimation and validation of Cryo-EM models.

**Figure supplement 2.** Assessment of Cryo-EM map quality and model fitting in the map.

The basic amine of the setron is in a bicyclic ring in granisetron and palonosetron, and a diazole ring in ondansetron and alosetron. The amine is within 4 Å of Trp156 (Loop B), Tyr207 (Loop C), Trp63 (Loop D) and Tyr126 (Loop E), and is likely to be involved in polar interactions with these residues. In particular, the carbonyl oxygen of Trp156 is close to the amine group of setron, and in the 5-HT$_{3A}$R-Alo, it forms a hydrogen bond with the amine group in the diazole ring. The relative orientation of the tertiary nitrogen and Trp156 is conducive for a cation-pi interaction, as seen in the AChBP-5-HT$_3$ chimera structure (*Kesters et al., 2013*). A similar interaction is also predicted for the primary amine group of serotonin (*Beene et al., 2002*). The aromatic end of the molecule is an

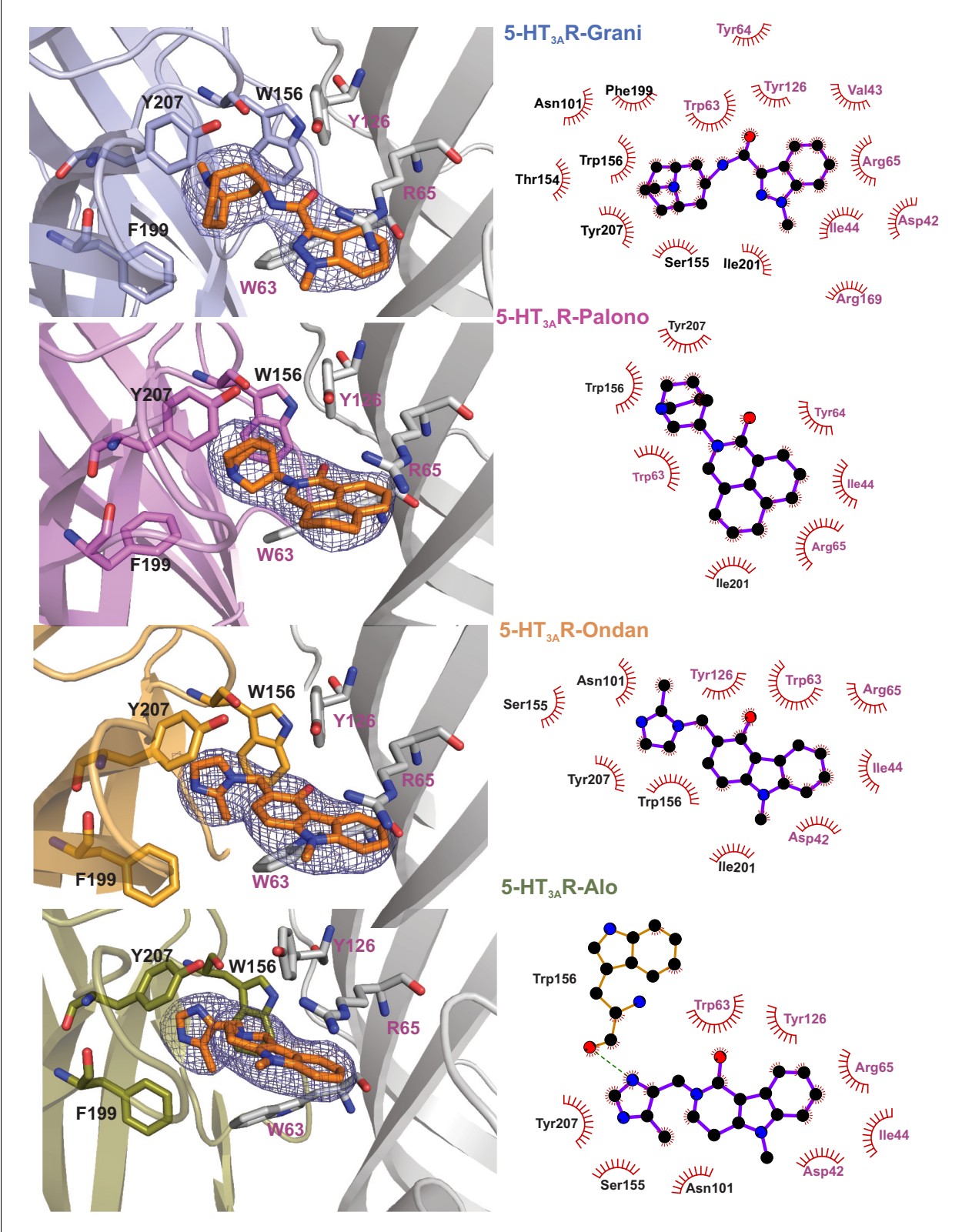

**Figure 2.** Setron-binding poses. (a) Cryo-EM density for the setrons, located at the canonical neurotransmitter-binding site. The map is contoured at $9\sigma$ (5-HT$_{3A}$R-Grani) (*Basak et al., 2019*); $8.5\sigma$ (5-HT$_{3A}$R-Palono); $7\sigma$ (5-HT$_{3A}$R-Ondan); $6\sigma$ (5-HT$_{3A}$R-Alo). The binding site lies at the interface of the principal (colored) and the complementary (gray) subunits. The binding-site residues are shown in stick representation with residues from the principal subunit labeled in *black* and those from the complementary subunit in *magenta*. From *top to bottom*: 5-HT$_{3A}$R-Grani, 5-HT$_{3A}$R-Palono, 5-HT$_{3A}$R-Ondan, and 5-

*Figure 2 continued on next page*

*Figure 2 continued*

HT$_3$AR-Alo. (**b**) LigPlot analysis of setron-5-HT$_3$R interactions. Most interactions with the setron are hydrophobic in nature (shown by *red arch with spikes*). Putative hydrogen bond between Trp156 and alosetron is shown as a *green* dotted line.

The online version of this article includes the following figure supplement(s) for figure 2:

**Figure supplement 1.** Interaction of setrons with neighboring residues.

**Figure supplement 2.** Arrangement of residues lining the binding site as seen in 5-HT$_3$R-Alo.

indazole in granisetron, isoquinoline in palonosetron, carbazole in ondansetron, and pyridoindole in alosetron. It is oriented toward the complementary subunit, and lies parallel to the membrane. In this orientation, the aromatic moiety is stabilized by a number of hydrophobic interactions with Ile44, Trp63, Tyr64, Ile201, and Tyr126 (shown by gray color in surface representation *Figure 2—figure supplement 1*). The setron molecule is within 4–5 Å and potentially makes π-π interactions (edge-to-face or face-to-face) with Trp63, Tyr126, Trp156, and Tyr207. These interactions are also consistent with our MD simulations (discussed below). While most interactions with setrons observed in these structures are apolar in nature, it is to be noted that water molecules were not modeled into the structures. Interactions mediated through water molecules are inferred from MD simulations (discussed below). In addition, the planar aromatic rings lie beneath Arg65, and are in close proximity to the positively charged nitrogen in the guanidinium group of Arg65, revealing a potential cation-pi interaction (*Figure 2—figure supplement 2*). This interaction was also observed in the AChBP-5-HT$_3$ chimera (*Kesters et al., 2013*) and 5-HT$_3$AR-Grani structures (*Basak et al., 2019*). As previously noted in 5-HT$_3$AR-Grani, the setron position causes reorientation of Arg65 (β2 strand or Loop D) and Trp168 (β8-β9; Loop F) (*Basak et al., 2019*). Earlier reports also predicted large orientational differences for Trp168 when the binding site was occupied by agonist or antagonist (*Thompson et al., 2006*). In this position, Arg65 is in a network of interactions involving Asp42 (β1), Try126 (β6), Trp168 (β8-β9; Loop F), Arg169 (β8-β9; Loop F), and Asp177 (β8-β9; Loop F) (*Figure 2—figure supplement 2b*). Glu102 (Loop A) which is in the vicinity of the ligand-binding site is in a hydrogen-bond network with Thr133 and Ala134 carbonyl (β6 strand). Interestingly, both of these networks are also present in serotonin-bound 5-HT$_3$AR, but absent in 5-HT$_3$AR-Apo, indicating the ligand-induced formation of the interaction network (*Basak et al., 2018a*; *Basak et al., 2018b*).

To understand the dynamics of ligand-receptor interactions, 100 ns MD simulations were carried out for 5-HT$_3$AR-Grani, 5-HT$_3$AR-Palono, 5-HT$_3$AR-Ondan and 5-HT$_3$AR-Alo, structures embedded in a 1-palmitoyl-2-oleoyl phosphatidyl choline (POPC) membrane and encased in water with 150 mM NaCl. The analysis also included simulations of the un-liganded (5-HT$_3$AR-Apo) and the serotonin-bound structures under the same conditions. In the presence of serotonin, two conformational states were resolved for 5-HT$_3$AR by previous cryo-EM studies; one was partially open (referred to State 1) and the other was open (referred to State 2) (*Basak et al., 2018b*). Although the two states had major differences in the TMD and the ICD, they were identical at the level of the serotonin-binding site and Loop C orientation. Given the better resolution of State 1 (referred to as 5-HT$_3$AR-Serotonin here and throughout), we used this structural model for comparison with setron-bound structures. While the cryo-EM density for the setrons allowed precise orientation of the ligand in the pocket, accurate modeling of the serotonin was limited by the cryo-EM resolution in combination with the smaller size of the molecule (*Basak et al., 2018b*). Upon evaluating the various docking poses for serotonin acquired using an initial pose placement with GlideSP followed by an in place refinement with GlideXP (Schrödinger Release 2019–2: Glide, Schrödinger, LLC, New York, NY, 2019), we found that the identified top-scored pose was essentially identical to that in the cryo-EM model.

To assess the stability of each ligand binding-pose modeled from cryo-EM density, we quantified the root mean square deviation (RMSD) of each pose relative to its starting conformation, assessed for each subunit independently every 500 ps of each ligand-5HT$_3$AR 100 ns simulation for a total of 200 simulation frames (*Figure 3a*). We also quantified the average RMSD of each pose relative to its starting conformation by averaging over 1000 simulation snapshots (200 frames sampled every 500 ps for each of the five subunits treated as replicates) extracted from the 100 ns simulations for each ligand-5HT$_3$AR complex. Among the ligands studied, the serotonin molecule exhibited considerable fluctuations (RMSD up to 5.4 Å with an average value of 1.9 Å, across five subunits) within the binding pocket. In contrast, most setron molecules maintained a low RMSD (RMSD average values of 1.1

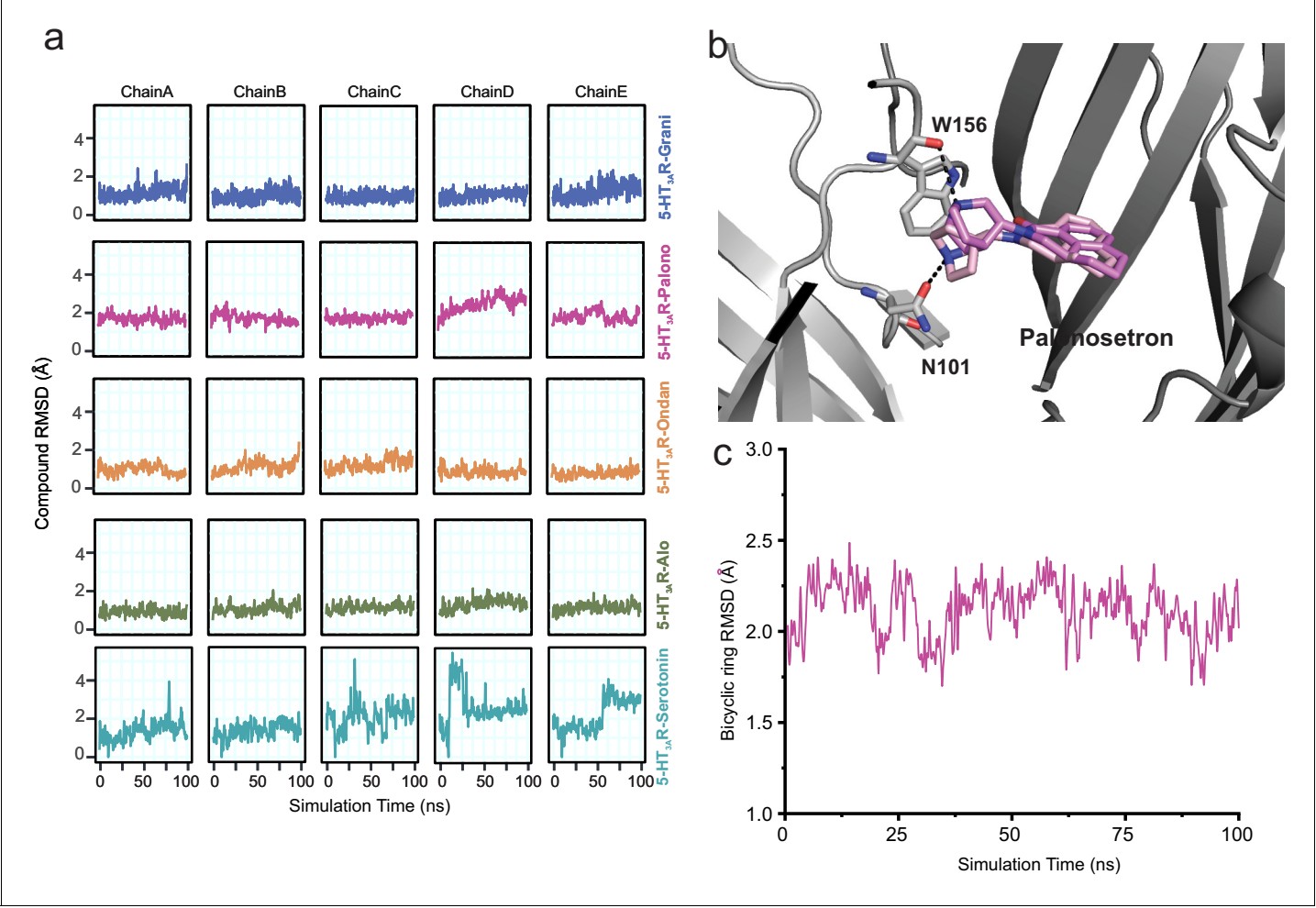

**Figure 3.** Assessment of conformation stability of ligand-binding poses by molecular dynamic simulations. (**a**) Time evolution of root mean square deviation (RMSD) of setrons' and serotonin's heavy atoms relative to their initial cryo-EM conformations of 5-HT$_{3A}$R for each protomer subunit. (**b**) Representative views of various palonosetron orientations during the 100 ns simulation. When the tertiary amine nitrogen in the bicyclic ring is pointing up, it interacts with the carbonyl oxygen of Trp156 and when it points down, it interacts with carbonyl oxygen of Asn101 side chain. (**c**) Time evolution of Root Mean Square Deviation (RMSD) of the bicyclic ring to its initial cryo-EM position.

The online version of this article includes the following figure supplement(s) for figure 3:

**Figure supplement 1.** Alignment of 5-HT$_{3A}$R-Palono with PDBID: 6Y1Z.

**Figure supplement 2.** Interaction Fingerprints in 5-HT$_{3A}$R-setrons and 5-HT$_{3A}$R-serotonin structures.

Å, 1.0 Å, and 1.1 Å for granisetron, ondansetron, and alosetron, respectively), with palonosetron demonstrating the largest RMSD (1.9 Å) among all the simulated setrons (*Figure 3a*). During the simulation, the palonosetron's bicyclic ring displayed fluctuation and positional reorientation. In these orientations, palonosetron had distinct interactions with binding-site residues, in particular with Asn101 in the 'down' position and Trp156 in the 'up' position (*Figure 3b and c*). A recently reported structure of 5-HT$_{3A}$R in complex with palonosetron (published during revision of this manuscript) shows an overall similar conformation of the protein as seen here (*Figure 3—figure supplement 1*; *Zarkadas et al., 2020*). Interestingly, the bicyclic ring orientation reported in this structure is similar to that captured in our MD simulations.

To evaluate the types of interactions that these ligands maintained with protein residues during MD simulation, we calculated 5-HT$_{3A}$R-ligand interaction fingerprints (see figure legend or methods for full interaction type definitions) averaged across each protomer in the complex (*Figure 3—figure supplement 2*). In the 5-HT$_{3A}$R-Serotonin simulations, the indole ring of serotonin makes pi-pi stacking interactions (edge-to-face or face-to-face) with key aromatic residues Trp63, Tyr126, and Trp156,

and Tyr207. Among these only interactions with Trp156 and Tyr126 occur with probability larger than 50%. Mutations to each of these aromatic position affects serotonin-binding and is reflected in increase in $EC_{50}$ for activation (*Beene et al., 2002*; *Spier and Lummis, 2000*; *Beene et al., 2004*). The indole nitrogen of serotonin forms water-mediated interactions with Asp42 and Arg169, albeit with a 25% probability whereas the ligand's hydroxyl group forms a hydrogen-bond with the amine nitrogen of Trp156 with a 50% probability. The indole nitrogen is also involved in an extended water network with Glu173 on Loop F, particularly in simulation frames where Loop F orients toward serotonin where this network is sampled as a two water -mediated hydrogen bond interaction. The primary amine nitrogen in serotonin makes multiple water-mediated interactions with the side chains of Glu209 and Thr154, and occasionally with Asn101 (less than 25% of the simulation time). In addition, serotonin forms relatively stable apolar interactions (>50% probability) with Ile44, Phe199, and Ile201.

The structural fingerprint analysis shows that all setrons form a number of very stable (probability >75%), mostly apolar, interactions with the following residues: Asp42, Val43, Ile44, Trp63, Tyr64, Arg65, Tyr126, and Trp156. While interactions between serotonin and residues Ile44, Trp63, Arg65, Tyr126, and Trp156 occurred with similar probability, those with Val43 and Tyr64 did not form at all and those with Asp42 were reduced, albeit complemented by hydrogen bond and water-mediated interactions. In 5-HT$_{3A}$R-Grani, the bicyclic nitrogen makes water-mediated hydrogen bonds with the sidechains of Thr154, Ser155, and Glu209. The indazole nitrogen occasionally interacts with Asp202 through a water-mediated hydrogen bond. Reduced affinity for granisetron is noted upon mutations at these positions (*Yan et al., 1999*) and notably W63A, W156A, and Y207A do not bind granisetron. In 5-HT$_{3A}$R-Palono, the tertiary amine on the ligand's bicyclic ring makes water-mediated and direct polar contacts with the backbone oxygen of Trp156 or the sidechain of Asn101 depending on the orientation of bicyclic ring. The carbonyl group of the isoquinoline moiety interacts with the amine nitrogen of Trp156 and carbonyl oxygen of Tyr64 through a water molecule. Mutations of Trp156 or Tyr64 cause large or small increases of palonosetron-induced inhibition, respectively (*Price et al., 2016*; *Del Cadia et al., 2013*; *Beene et al., 2002*). Interestingly, while N101Q preserves palonosetron-induced inhibition similar to wild-type receptor, a mutation to N101A (no H-bond with side chain) increases the potency of palonosetron (*Price et al., 2016*). Palonosetron forms an apolar interaction with Arg169 with higher probability compared to the other ligands, potentially due to palonosetron's pose forming weak interactions with loop C in favor of loop F. In 5-HT$_{3A}$R-Ondan, the ondansetron molecule forms highly probable edge-to-face stacking interactions with Tyr126 and Tyr207. The secondary nitrogen on the diazole ring forms a hydrogen-bond interaction with Glu209 through a water molecule. These water-mediated interactions are also occasionally seen with Thr154 and Asn101. In 5-HT$_{3A}$R-Alo, one of the secondary amine nitrogen on the diazole ring forms a water-mediated hydrogen-bond interaction with the carbonyl oxygen of Trp156. The amide oxygen of Trp156 forms a water-mediated hydrogen-bond contact with the amine group of the ligand's imidazole. The backbone oxygen of Tyr64 interacts with the carbonyl group of the ligand's pyridoindole ring through a water molecule. Interestingly, water-mediated interactions with Glu209 are absent in the alosetron fingerprint when compared to the ondansetron, granisetron, and serotonin fingerprints.

It is to be noted that many of the ligand-receptor interactions identified as important by the MD simulations are not directly evident from the Cryo-EM structures particularly since many of these interactions are mediated through water molecules (not modeled in the structural coordinates). In addition, MD simulation captures several transient interactions arising from side-chain flexibility and drug mobility within the pocket.

## Conformation of loop C

We previously showed that instead of being in a 5-HT$_{3A}$R-Apo like conformation, 5-HT$_{3A}$R-Grani revealed a counter-clockwise twist of beta strands in the ECD leading to a small inward movement of Loop C (connecting β9-β10 strands) closing-in on granisetron (*Basak et al., 2019*). The Loop C conformation has been correlated to the agonistic nature of the ligand in the binding site. The AChBP-ligand complexes have shown that agonist binding induces a 'closure' of Loop C, capping the ligand-binding site (*Hansen et al., 2005*). The 5-HT$_{3A}$R and other pLGIC structures solved thus far, in the apo and agonist-bound states, also follow this general trend (*Polovinkin et al., 2018*; *Basak et al., 2018a*; *Basak et al., 2018b*; *Du et al., 2015*; *Kumar et al., 2020*). This conformational

change may be part of a conserved pLGIC mechanism that couples ligand binding to channel opening through the ECD-TMD interfacial loops. However, studies have shown that unliganded pLGIC gating kinetics remain unaffected by Loop C truncation (*Purohit and Auerbach, 2013*), raising ambiguity over the role of Loop C closure in channel opening. Antagonist-bound AChBP structures show that Loop C further extended outward (*Hansen et al., 2005*), while partial agonists seem to induce partial Loop C closure but not to the level achieved by agonists (*Hibbs et al., 2009*), suggesting a correlation between the degree of Loop C closure and the level of agonism. However, in the crystal structure of AChBP in complex with antagonist dihydro-$\beta$-erythroidine, Loop C appears to move inward (*Shahsavar et al., 2012*).

In comparison to the 5-HT$_{3A}$R-Apo structure, Loop C adopts varying degrees of an inward conformation, and in the 5-HT$_{3A}$R-Alo structure the orientation is similar to 5-HT$_{3A}$R-serotonin (*Basak et al., 2019*; *Figure 4a*). The twisting inward movement does not pertain to Loop C alone, but it is also shared by adjoining β7, β9 and β10 strands forming the outer-sheets of the β-sandwich core, with notable deviations from the corresponding regions in the 5-HT$_{3A}$R-Apo structure (*Figure 4b*). In contrast, only minimal changes are observed in the β-strands of the inner sheets (β1, β2, β6) (*Figure 4b and c*). These conformational changes approach those seen in the 5-HT$_{3A}$R-Serotonin structure (*Basak et al., 2018b*). Although these results appear to diverge from the classical view that competitive antagonists either cause steric hindrance to agonist binding or evoke structural changes that are opposite to those caused by agonist, they are generally consistent with previous findings from dynamics studies in pLGIC using voltage-clamp fluorometry (VCF). In VCF, ligand-induced conformational changes and channel function are simultaneously monitored. When reporter-groups were introduced in Loop C of 5-HT$_{3A}$R, similar fluorescence changes were recorded from binding either serotonin or setrons (*Munro et al., 2019*). In GlyR, both glycine (agonist) and strychnine (a competitive antagonist) produced identical fluorescence responses from labels in certain regions of Loop C (*Pless and Lynch, 2009*), and a similar trend was also observed in ρ1GABAR (*Chang and Weiss, 2002*). These findings implied that the local structural changes induced by these ligands were indiscriminate to the functional response from the channel. Overall, these findings underscore the complexity of Loop C movement and its role in coupling channel opening. Interestingly, the inner β-sheet regions (particularly Loop E, contributing to the binding pocket from the complementary subunit) undergo distinct movements depending on the nature of the bound ligand in GlyR, 5-HT$_{3A}$R, and GABA$_A$R (*Munro et al., 2019*; *Pless and Lynch, 2009*; *Chang and Weiss, 2002*; *Muroi et al., 2006*) suggesting that this region maybe a better reporter for ligand discrimination.

Some of the differences in ligand-receptor interaction fingerprints across the different systems, particularly those involving residues Tyr207, Phe199, and Glu209 arise from differential positioning of Loop C. To characterize the flexibility of Loop C in the liganded and unliganded states of 5-HT$_{3A}$R, we monitored two structural parameters during the MD simulation runs. First, we assessed the RMSD of Loop C for each protomer in each 5-HT$_{3A}$R simulation by evaluating the distances of Cα, carbonyl carbon, and backbone nitrogen atoms of residues Ser200 through Asn205 with respect to their initial cryo-EM conformations (*Figure 4—figure supplement 1a*). Second, we defined a custom dihedral formed by the Cα atoms of residues Ala208, Phe199, Glu198, and Ile203 that measured the orientation of the loop relative to the complementary subunit binding site (*Figure 4—figure supplement 1b*). This dihedral was defined in such a way that a large angle would denote that the loop is oriented away from the binding site and small or negative angles indicate that the loop is oriented toward the binding site. Comparing these two parameters across the different systems suggest that Loop C is stable in its initial cryo-EM conformation in the case of 5-HT$_{3A}$R-Alo and 5-HT$_{3A}$R-Serotonin simulations where it predominantly adopts a 'closed' conformation. In other setron complexes, Loop C is observed to switch to an alternate 'open' conformation where it extends away from the binding-site surface. In comparison to all the ligand-bound structures, Loop C exhibited a much larger flexibility in 5-HT$_{3A}$R-Apo as evidenced by large RMSD values and a wide-range of dihedral angles. These multiple Loop C 'opening' and 'closing' events in the 5-HT$_{3A}$R-Apo structure contrasts ligand-bound states of the channel and suggest that the presence of a ligand in the binding-site forces the loop C to remain 'closed'. Such enhanced flexibility in the unliganded state of 5-HT$_{3A}$R has also been reported in a previous 20 μs simulation study (*Guros et al., 2020*).

To assess the impact of Loop C movement on the relative size of each setron-binding pocket, we quantified the average number of water molecules found within each binding site assessed

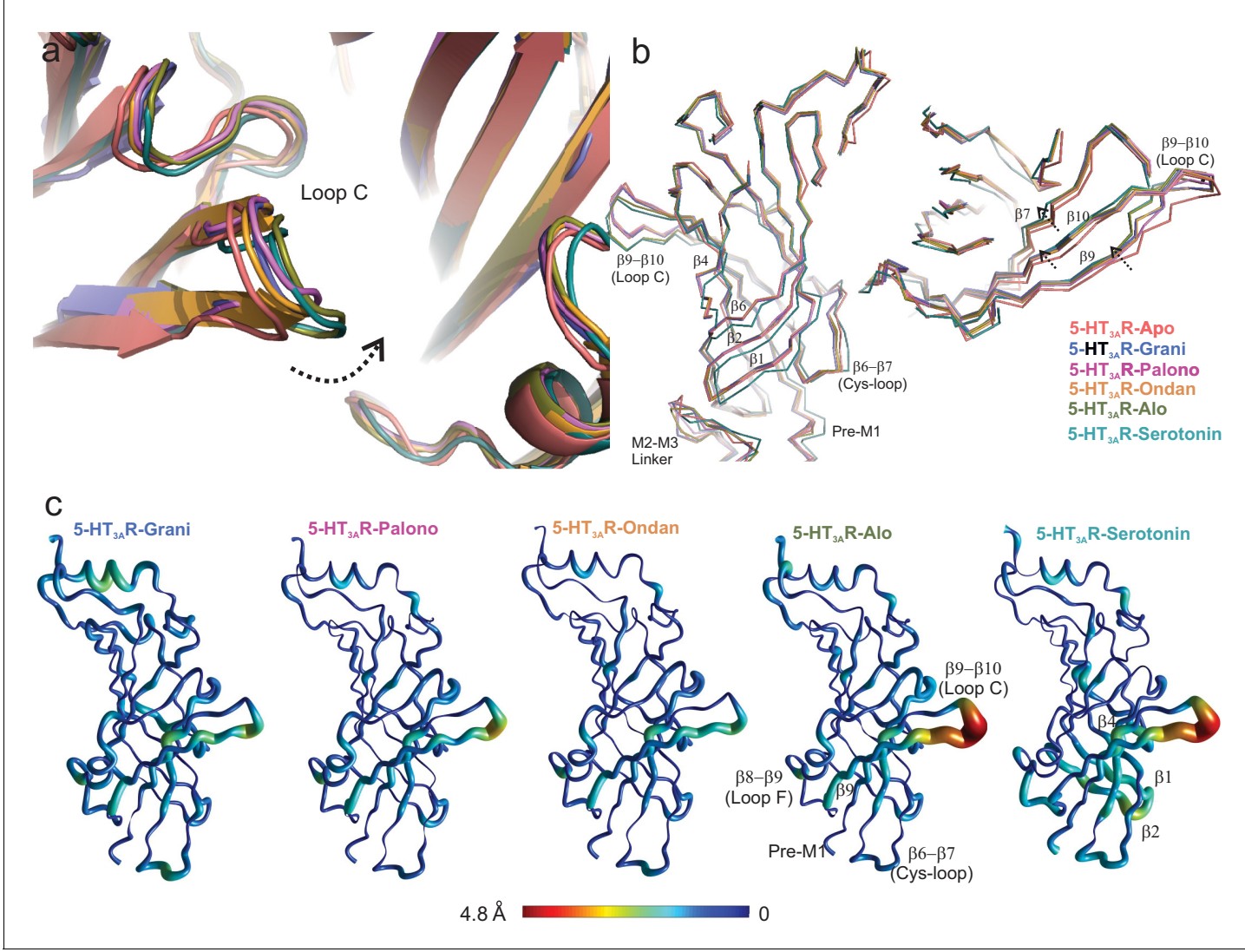

**Figure 4.** Setron-binding pocket and conformational changes in Loop C. (a) Global alignment of 5-HT$_{3A}$R-Apo, 5-HT$_{3A}$R-State1 (serotonin-bound), 5-HT$_{3A}$R-Grani, 5-HT$_{3A}$R-Palono, 5-HT$_{3A}$R-Ondan, and 5-HT$_{3A}$R-Alo structures. With respect to 5-HT$_{3A}$R-Apo, the serotonin- and setron- bound conformations reveal an inward positioning of Loop C (*shown by arrow*). (b) Relative displacement of the inner β-strands seen from a side-view (*left panel*) and the outer β-strands seen from the top (*right panel*). Arrows indicate the direction of movement. (c) Pentameric assembly of setron- and serotonin-bound structures were aligned to Apo-5-HT$_{3A}$R. A cubic spline interpolation was then done to smoothly connect cα displacement for each structure and mapped by short cylinders, whose diameters are equivalent to the displacement at that position compared to Apo-5-HT$_{3A}$R. The color was also scaled to the same value using the color map shown. The analysis was done in Matlab v2019a (Mathworks, Natick MA).
The online version of this article includes the following figure supplement(s) for figure 4:

**Figure supplement 1.** Conformational differences in Loop C.

separately for each protomer chain. This was evaluated by counting water oxygen atoms within 3 Å of any ligand atom. Since each ligand maintained its overall binding mode during simulation, these measurements represent an approximation of binding-site volume. These data show that alosetron, ondansetron, and serotonin have a relatively lower number of water molecules within their binding sites (*Figure 5*).

To further investigate the motion of Loop C between the 'closed' cryo-EM structure and the MD sampled 'open' conformation we evaluated the minimum polar side-chain atom distance between Arg65 and Asp202, residues known to form a hydrogen-bond interaction that may effectively rigidify Loop C in a 'closed' conformation (*Guros et al., 2020*). We hypothesized that in our MD simulations Loop C would not adopt an 'open' conformation if an Arg65-Asp202 interaction was formed. We

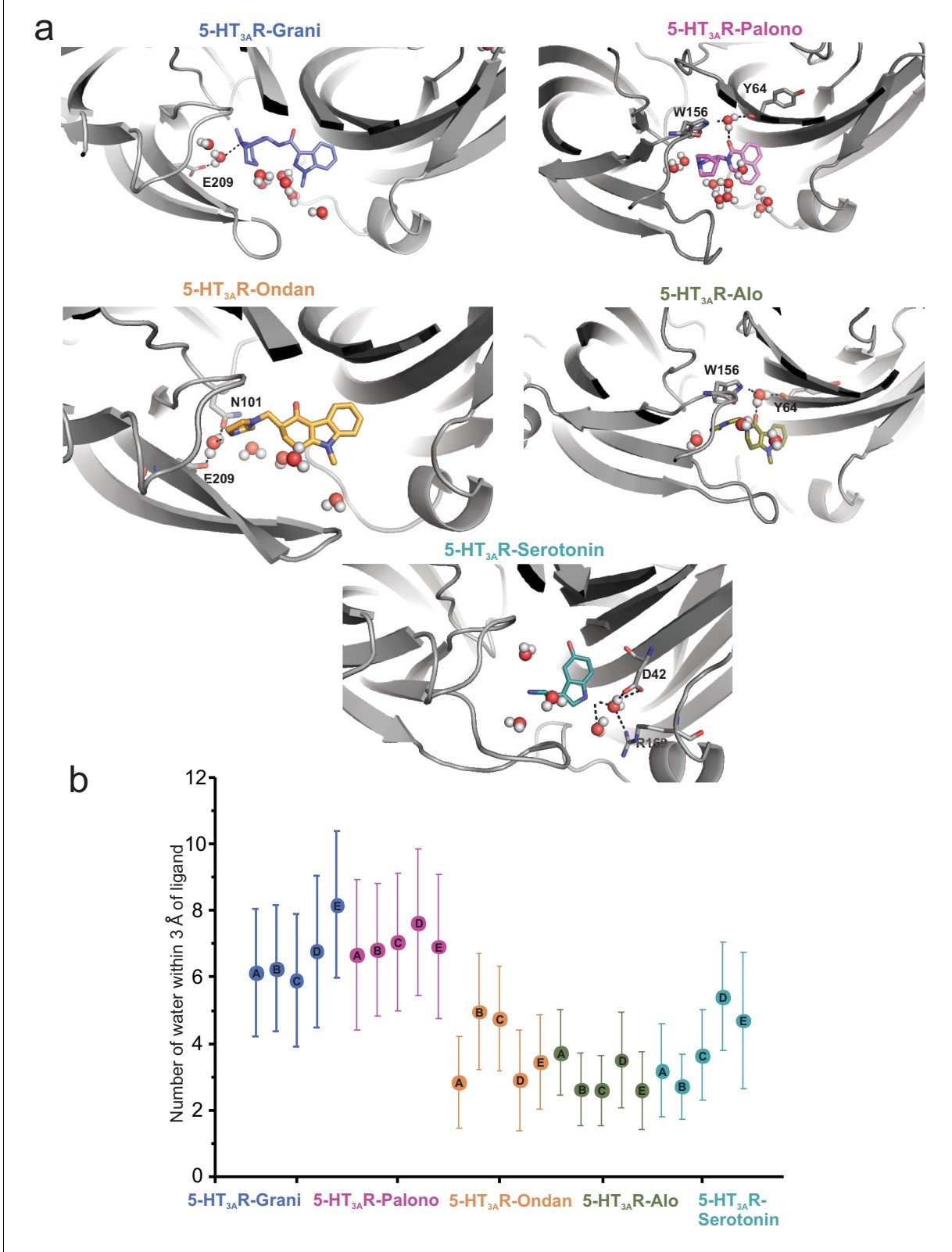

**Figure 5.** Assessment of the number of water molecules present within each ligand-binding site during MD simulation. (a) Snapshots during the simulation showing water molecules in the pocket. (b) Average number of water molecules (defined as a count of water oxygen atoms within 3 Å of any setron atoms) for each setron- and serotonin-bound simulation subdivided by protomer and the corresponding standard deviation.

find that the 5-HT$_{3A}$R-Alo and 5-HT$_{3A}$R-Serotonin MD simulations maintained an interaction between Arg65 and Asp202 more often than in any other setron-bound structure, and that most setron-bound simulations did not appreciably form this stabilizing interaction (*Figure 6a and b*). Thus, our mechanistic hypothesis is such that when Arg65 is interacting with Asp202, Loop C is in a stable 'closed' conformation, which in turn reduces the accessibility of the binding pocket to water, and incidentally contributes to the higher stability of the ligand binding pose.

Our MD simulations suggest that Arg65 may have a differential effect on the binding of various setrons. In agreement, mutations at the Arg65 position in human 5-HT$_{3A}$R abolish granisetron binding but tropisetron binding is only reduced (*Ruepp et al., 2017*). To further assess the role of Arg65 in binding various setrons, we measured the extent of inhibition of serotonin-induced currents. Since for competitive antagonists the extent of inhibition depends on agonist concentration, the serotonin concentration in each case was kept close to the EC$_{50}$ value for wild type (2 μM) and R65A (10 μM) (*Figure 6c*). Granisetron and palonosetron inhibition was measured at 1 nM; ondansetron and alosetron inhibition was measured at 0.1 nM (these concentrations were chosen to achieve a 50% inhibition for wild type upon co-application with serotonin) (*Figure 6d and e*). Of note, co-application of setron in some cases has ~100 fold lower effect than pre-application due to slow on-rates (*Lummis and Thompson, 2013*). Mutational perturbation at Arg65 has a significant effect on inhibition by each setron, albeit to varying extents. While revealing a functional effect on serotonin and setron binding, the R65A mutational studies do not provide conclusive evidence for differential effects of various setrons. We think that R65 plays a role in concert with neighboring residues. Additional mutagenesis and combination of mutations may be needed to understand the proposed mechanism better.

## Conformational differences along the ion permeation pathway

Analysis of the ion permeation pathway along the pore axis shows a slight constriction in the middle of the ECD lined by residues Lys108 and Asp105 in the β4-β5 loop. The Asp105 position is conserved among most cation-selective pLGICs and mutations at this position affect single-channel conductance and open probability in pLGICs (*Livesey et al., 2011*; *Sine et al., 2010*; *Chakrapani et al., 2003*). The ECD constriction is narrower in the 5-HT$_{3A}$R-Apo structure and widens in the serotonin-bound structures. 5-HT$_{3A}$R-setron structures show varying extents of widening at this position. However, previous studies assessing permeation of water molecules and of ions with imposed membrane potential have shown that this constriction point does not impede ion permeation in the 5-HT$_{3A}$R-Apo and 5-HT$_{3A}$R-Serotonin structures (*Basak et al., 2018b*). Conformational changes are also present in the TMD and may arise from small twisting movements in the ECD. Interestingly, in each of the 5-HT$_{3A}$R-setrons structures, the pore-lining M2 helices are positioned away from the central axis, and are in a more-expanded conformation than in the 5-HT$_{3A}$R-Apo structure (*Figure 7a*). At positions Val264 (13'), Leu260 (9'), Ser253 (2'), and Glu250 (−1'), the pore radii in 5-HT$_{3A}$R-setron structures are larger than in the 5-HT$_{3A}$R-Apo structure. However, in all these structures, the pore is constricted to below the hydrated Na$^+$ radii (*Marcus, 1988*; *Figure 7b*).

Local dynamics of the permeation pathway were monitored during the aforementioned 100 ns MD simulations of 5-HT$_{3A}$R-Apo, 5-HT$_{3A}$R-Serotonin, and each of the 5-HT$_{3A}$R-setron structures embedded in a POPC membrane encased in water and 150 mM NaCl (*Figure 7—figure supplement 1*). As expected, no major changes to the overall pore profile of the channel were observed during simulations, which were carried out in the absence of positional restraints. The side-chain movements of the residues lining the permeation pathway caused only slight fluctuations of the pore radius as shown by the standard deviations from across eight equidistant simulation frames for each system. Most notably, in each case, the pore remained constricted at the Leu260 (9') to under 2.3 Å (below the hydrated Na$^+$ radius). As also reported in earlier simulation studies, the Leu260 (9') position is the major barrier in the TMD to ion permeation in the 5-HT$_{3A}$R-Apo and 5-HT$_{3A}$R-Serotonin structures (*Basak et al., 2018b*). While there are small conformational changes in the ICD, the post-M3 loop occludes the lateral portals at the interface of the TMD and ICD which are predicted to be the ion exit paths. The extent of occlusion is similar to that seen in the 5-HT$_{3A}$R-Apo structure (*Basak et al., 2018a*) suggesting that the ICD exits are closed in these conformations. Interestingly, in muscle-type nAChR, the ICD portals appear to be open even in the Apo conformation, highlighting the mechanistic differences among pLGIC members (*Rahman et al., 2020*). Overall, these analyses suggest that although there are different extents of conformational changes in the 5-HT$_{3A}$R-

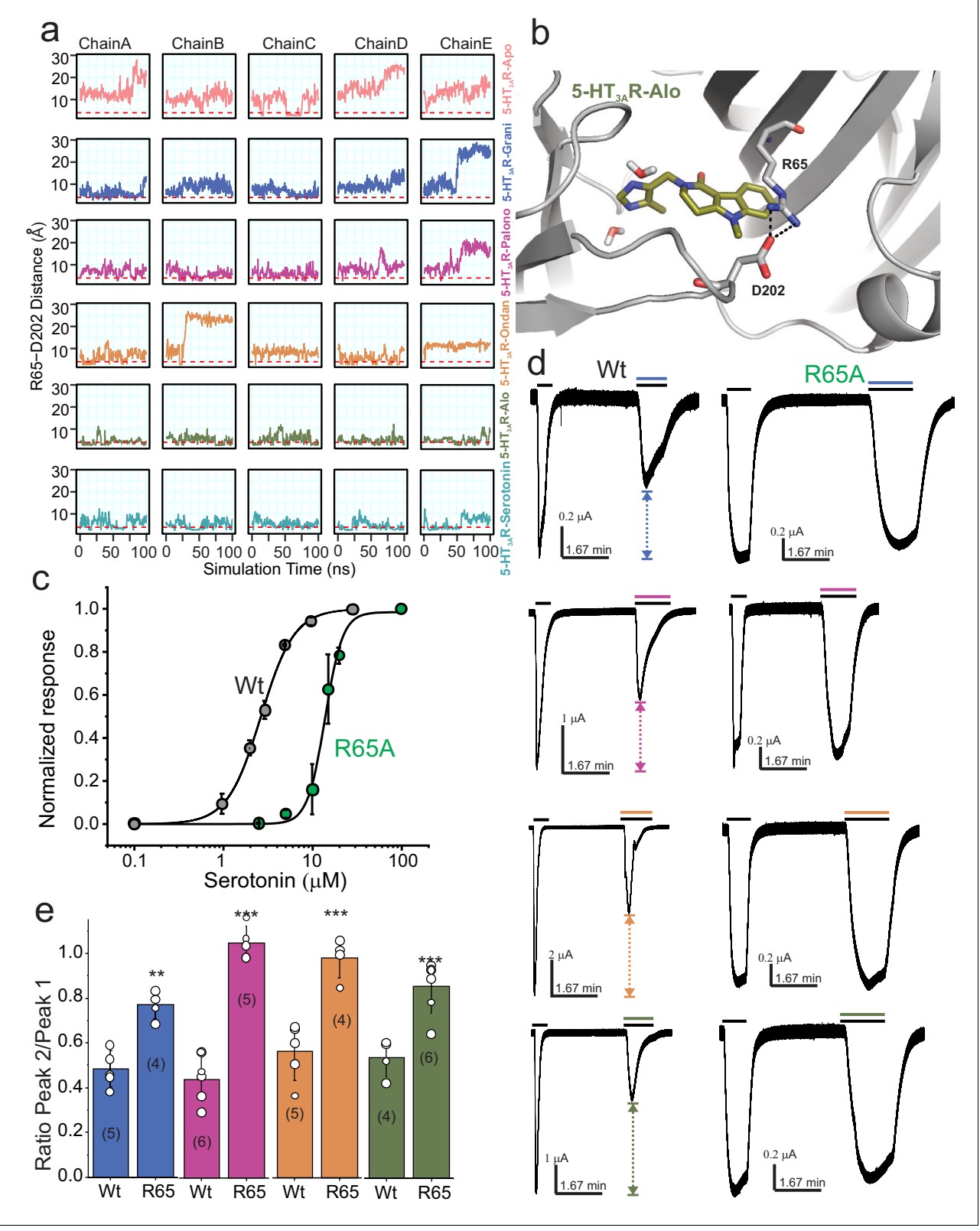

**Figure 6.** Dynamic interaction between Arg65 and Asp202. (a) Time evolution of the minimum distance between side-chain polar atoms of Arg65 and Asp202 throughout 100 ns simulations. A 4 Å distance threshold is shown as a red dashed line to denote a generous cutoff for H-bond interactions between these residues. (b) MD snapshot that show the Arg65-Asp202 interaction. (c) Dose-response curve for serotonin activation measured by TEVC recordings (at −60 mV) for WT 5-HT$_{3A}$R and R65A expressed in oocytes. The EC$_{50}$, the Hill coefficient (nH), and the number of independent oocyte

*Figure 6 continued on next page*

*Figure 6 continued*

experiments are: WT (EC$_{50}$: 2.70 ± 0.09 µM; nH: 2.3 ± 0.17; n: 3) and R65A (EC$_{50}$: 13.79 ± 0.50 µM; nH: 4.4 ± 0.59; n: 4) (*Basak et al., 2019*) (**d**) Functional analysis of Arg65. Currents were elicited in response to serotonin (concentrations used near EC$_{50}$ values WT- 2 µM, and R65A- 10 µM) with and without co-application of setrons. Dotted arrows show the extent of setron inhibition in each case. (**e**) A plot of the ratio of peak current in the presence of setron to peak current in the absence of setron is shown for WT and R65A. Data are shown as mean ± s.d (n is indicated in parenthesis). Significance at p=0.01 (***) and p=0.05 (**) calculated by two sample t-test for wild type and R65A.

The online version of this article includes the following source data for figure 6:

**Source data 1.** Source Data for *Figure 6c and e*.

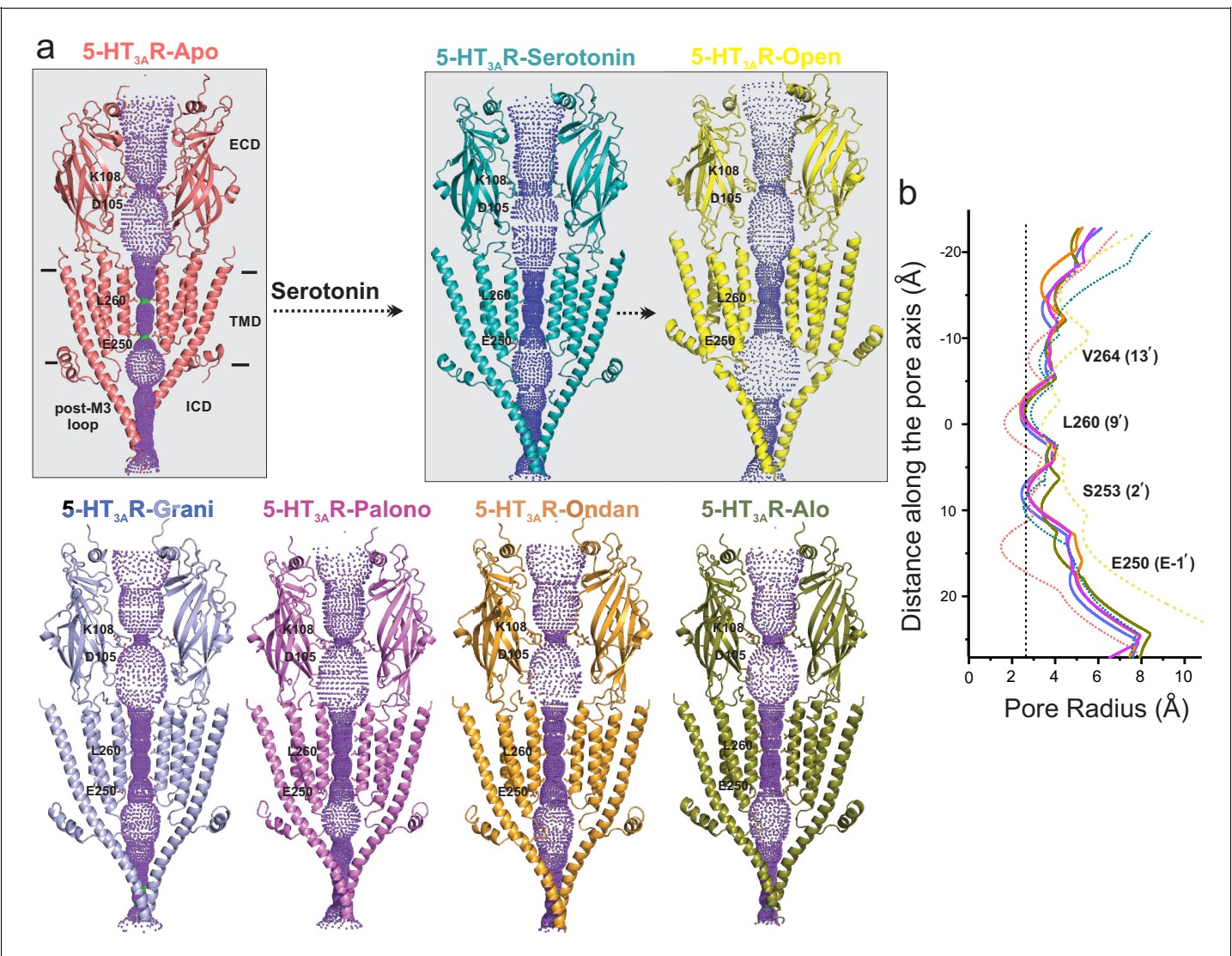

**Figure 7.** Pore profiles of 5-HT$_{3A}$R in Apo, serotonin-, and setron- bound states. (**a**) Ion conduction pathway predicted by HOLE (*Smart et al., 1996*). Models are shown in cartoon representation. Only two subunits are shown for clarity. The locations of pore constrictions are shown as sticks. (**b**) The pore radius is plotted as a function of distance along the pore axis. The dotted line indicates the approximate radius of a hydrated Na$^+$ ion which is estimated at 2.76 Å (right) (*Marcus, 1988*).

The online version of this article includes the following figure supplement(s) for figure 7:

**Figure supplement 1.** Assessment of pore radii of 5-HT$_{3A}$R structures in the Apo, setron-, and serotonin-bound states by MD simulations.
**Figure supplement 2.** Multiple sequence alignment of 5-HT$_3$R.

setron structures that lie between those of 5-HT$_{3A}$R-Apo and 5-HT$_{3A}$R-Serotonin, each of these structures appears to be non-conducting to ions.

A limitation of these standard, shorter MD simulation timescales is that dynamic transitions between multiple states or allostery between ligand-binding at the ECD and pore-opening at the TMD are not expected to be captured during runs. The purpose of these simulations was to verify ligand stability in the pocket and the overall stability of the cryo-EM conformation rather than ligand-induced conformational rearrangements. Future studies with enhanced MD simulations may provide insights into transitions between different conformational states and the mechanistic details of coupling across domains.

### Setron-binding sites in heteromeric 5-HT$_3$R

We further explored the question of whether setrons discriminate between homomeric 5-HT$_{3A}$R and the heteromeric assemblies of subunit A in combination with either B, C, D, or E subunits. Among the different heteromeric 5-HT$_3$R assemblies, the most studied is the 5-HT$_{3AB}$R. A sequence alignment of mouse and human 5-HT$_3$R subunits (*Figure 7—figure supplement 2*) shows that two key residues that interact with serotonin and setrons, Trp156 (on the principal side) and Arg65 (on the complementary side), are present exclusively in subunit A. This suggests that the setron- binding site may be limited to A-A interfaces both in the 5-HT$_{3A}$R homomeric and heteromeric assemblies. A similar conclusion has also been drawn from earlier studies that have shown that setrons do not significantly differ in potency between 5-HT$_{3A}$R and 5-HT$_{3AB}$R, and that mutations to binding-site residues in subunit A had more dramatic effects on antagonist-binding affinity and an increased serotonin EC$_{50}$ than mutations to equivalent positions in subunit B (*Del Cadia et al., 2013*; *Lochner and Lummis, 2010*).

### Summary

A comprehensive structural analysis of multiple high-resolution structures of setron-bound 5- HT$_{3A}$R complexes reveal several features of competitive antagonism that were not fully evident from the previous structural findings. Serotonin binds within a partially solvent-exposed cavity at the subunit interface and elicits Loop C closure and twisting of the β-strands within the ECD. The setron-binding pocket, while involving overlapping residues, extends further into the complementary subunit. Setron-binding evokes varying degrees of Loop C closure and in some cases, almost to the same degree as in the serotonin-bound state. The Loop C movements are associated with varying degrees of structural changes in the inner and outer β-strands that translate to small changes in the pore-lining M2 helices. Overall, setrons stabilize 5-HT$_{3A}$R conformational states that are non-conductive, but appear to lie between the apo and serotonin-bound states. These findings therefore suggest that competitive antagonism in 5-HT$_{3A}$R, and potentially in other pLGIC, may involve stabilizing intermediates along the activation pathway. With new emerging uses of setrons to treat psychiatric disorders, inflammation, substance abuse, and Alzheimer's disease, these studies lay the foundation for the design of novel therapeutics that may have higher treatment efficacy and potentially fewer off-target effects.

## Materials and methods

### Electrophysiological measurements in oocytes

Mouse 5-HT$_{3A}$R gene (purchased from GenScript) and mutant genes were inserted into pTLN plasmid. The plasmids were linearized with MluI restriction enzyme by digesting overnight at 37°C. The mMessage mMachine kit (Ambion) was used to make mRNA as per the manufacturer's protocol and cleanup using RNAeasy kit (Qiagen). 3–10 ng of mRNA was injected into *X. laevis* oocytes (stages V– VI), and incubated for 2–5 days, after which current recordings were performed. Water injected oocytes were used as a control to verify that no endogenous currents were present. Female *X. laevis* were purchased from Nasco and kindly provided by W. F. Boron. The Institutional Animal Care and Use Committee (IACUC) of Case Western Reserve University approved the animal experimental procedures. Oocytes were maintained in OR3 medium (GIBCO-BRL Leibovitz medium containing glutamate, 500 units each of penicillin and streptomycin, pH adjusted to 7.5, osmolarity adjusted to 197 mOsm) at 18°C. Warner Instruments Oocyte Clamp OC-725 was used to perform two-electrode

voltage-clamp experiments at a holding potential of −60 mV. Currents were sampled and digitized at 500 Hz with a Digidata 1332A. Clampfit 10.2 (Molecular Devices) was used to analyze experimental data. Perfusion solution consisted of 96 mM NaCl, 2 mM KCl, 1.8 mM CaCl$_2$, 1 mM MgCl$_2$, and 5 mM HEPES (pH 7.4, osmolarity adjusted to 195 mOsM) was used at a flow rate of 6 ml/min. Chemical reagents (serotonin hydrochloride, alosetron hydrochloride, ondansetron hydrochloride, and palonosetron hydrochloride) were purchased from Sigma-Aldrich.

## Full-length 5-HT$_{3A}$R cloning and transfection

The mouse 5-HT$_{3A}$R (NCBI Reference Sequence: NM_001099644.1) gene was codon-optimized for *Spodoptera frugiperda* (Sf9) cells and purchased from GenScript. The construct consists of the 5-HT$_{3A}$R gene along with a C-terminal 1D4-tag (*MacKenzie et al., 1984*) and four strep-tags (WSHPQFEK) at the N terminus, each separated by a linker sequence (GGGSGGGSGGGS) and followed by a TEV-cleavage sequence (ENLYFQG). Sf9 cells (Expression System) were grown in ESF921 medium (Expression Systems) at 28°C without CO$_2$ exchange and in absence of antibiotics. Cellfectin II reagent (Invitrogen) was used for transfection of recombinant 5-HT$_{3A}$R bacmid DNA into sub-confluent Sf9 cells. After 72 hr of transfection, the progeny 1 (P1) recombinant baculoviruses were obtained by collecting the cell culture supernatant. The P1 was then used to infect Sf9 cells which produced P2 viruses, and subsequently P3 viruses from the P2 virus stock. The P3 viruses were used for recombinant protein expression.

## 5-HT$_{3A}$R expression and purification

Sf9 cells were grown to approximately $2.5 \times 10^6$ per ml followed by infection with P3 viruses. After 72 hr post-infection, the cells were centrifuged at 8,000 g for 20 min at 4°C to separate the supernatant from the pellet. The cell pellet was resuspended in 20 mM Tris-HCl, pH 7.5, 36.5 mM sucrose supplemented with 1% protease inhibitor cocktail (Sigma-Aldrich). Cells were sonicated on ice. Non-lysed cells were pelleted down by centrifugation (3,000 g for 15 min) and the supernatant was collected. The membrane fraction was separated by ultracentrifugation (167,000 x g for 1 hr) and solubilized in 50 mM Tris pH 7.5, 500 mM NaCl, 10% glycerol, 0.5% protease inhibitor and 1% C12E9 for 2 hr at 4°C. Non-solubilized material was removed by ultracentrifugation (167,000 x g for 15 min). The solubilized membrane proteins containing 5-HT$_{3A}$ receptors were bound with 1D4 beads pre-equilibrated with 20 mM HEPES pH 8.0, 150 mM NaCl and 0.01% C12E9 for 2 hr at 4°C. The non-bound proteins were removed by washing beads with 100 column volumes of 20 mM HEPES pH 8.0, 150 mM NaCl, and 0.01% C12E9 (buffer A). The protein was then eluted with 3 mg/ml 1D4 peptide (TETSQVAPA) which is solubilized in buffer A. Eluted protein was deglycosylated with PNGase F (NEB) by incubating 5 units of the enzyme per 1 µg of protein for 2 hr at 37°C under gentle agitation. Deglycosylated protein was then purified using a Superose six column (GE healthcare) equilibrated with buffer A. Purified protein was concentrated to 2–3 mg/ml using 50 kDa MWCO Millipore filters (Amicon) for cryo-EM studies.

## Cryo-EM sample preparation and data acquisition

5-HT$_{3A}$R protein (~2.5 mg/ml) was filtered and incubated with 100 µM drugs (Alosetron, Ondansetron, and Palonosetron) for 1 hr. Fluorinated Fos-choline-8 (Anatrace) was added to the protein sample to a final concentration of 3 mM. The protein was then blotted onto Cu 300 mesh Quantifoil 1.2/1.3 grids (Quantifoil Micro Tools) two times with 3.5 µl sample each time, and the grids were plunge frozen immediately into liquid ethane using a Vitrobot (FEI). The grids were imaged using a 300 kV FEI Titan Krios G3i microscope equipped with a Gatan K3 direct electron detector camera. Movies containing ~50 frames were collected at 105,000 × magnification (set on microscope) in super-resolution mode with a physical pixel size of 0.848 Å/pixel, dose per frame 1 e$^-$/Å (*Gilmore et al., 2018*). Defocus values of the images ranged from −1.0 to −2.5 µm (input range setting for data collection) as per the automated imaging software SerialEM (*Mastronarde, 2005*).

## Image processing

MotionCor2 (*Zheng et al., 2017*) was used to correct beam-induced motion using a B-factor of 150 pixels (*Gilmore et al., 2018*). Super-resolution images were binned (2 × 2) in Fourier space, making a final pixel size of 0.848 Å. Entire data processing was conducted in RELION 3.1 (*Fernandez-*

*Leiro and Scheres, 2017*). CTF of the motion-corrected micrographs were estimated using Gctf software (*Mindell and Grigorieff, 2003*). Auto-picked particles from total micrographs (*Table 1*) from individual datasets (each drug) were subjected to 2D classification to remove suboptimal particles. An initial 3D reference model was generated from the 5-HT$_{3A}$R-apo cryo-EM structure (RCSB Protein Data Bank code (PDB ID): 6BE1). The model was low-pass filtered at 60 Å using EMAN2 (*Tang et al., 2007*). Iterative 3D classifications, 3D auto-refinements, and Bayesian polishing generated density model of Alosetron, Ondansetron and Palonosetron bound 5-HT$_{3A}$R with 42, 065 particles, 67, 333 particles, and 91,163 particles, respectively. During 3D classifications each of the classes was investigated carefully and particles appeared to belong to a single conformation. Per-particle contrast transfer function (CTF) refinement and beam tilt correction were applied followed by a final 3D-autorefinement. A soft mask was generated in RELION and used during the post-processing step, which resulted in an overall resolution of 3.32 Å, 3.06 Å, and 2.92 Å for, 5-HT$_{3A}$R-Palono, 5-HT$_{3A}$R-Ondan, and 5-HT$_{3A}$R-Alo respectively (calculated based on the gold-standard Fourier shell coefficient (FSC) = 0.143 criterion, *Table 1*). B-factor estimation and map sharpening were performed in the post-processing step in RELION. The ResMap program was used to calculate local resolutions (*Kucukelbir et al., 2014*).

**Table 1.** Cryo-EM data collection/processing.

| | 5-HT$_{3A}$-Alosetron (EMDB-21511; PDB-6W1J) | 5-HT$_{3A}$-Ondansetron (EMDB-21512; PDB-6W1M) | 5-HT$_{3A}$-Palonosetron (EMDB-21518; PDB-6W1Y) |
|---|---|---|---|
| Data collection and processing | | | |
| Magnification | 105,000x | | |
| Voltage (kV) | 300 | | |
| Data collection mode | Super-resolution | | |
| Electron exposure (e–/Å$^2$) | 50 | | |
| Defocus range (μm) | −1.2 to −2.5 | | |
| Physical Pixel size (Å/pixel) | 0.848 | | |
| Symmetry-imposed | C5 | | |
| Initial particle images (no.) | 568,452 | 449,628 | 1,114,542 |
| Final particle images (no.) | 42,065 | 67,333 | 91,163 |
| Map resolution (unmasked, Å) at FSC$_{143}$ | 3.2 | 3.4 | 3.7 |
| Map resolution (masked, Å) at FSC$_{143}$ | 2.92 | 3.06 | 3.32 |
| Map resolution range (Local resolution) | 2.5–4.5 | 2.5–4.5 | 2.5–4.5 |
| Refinement | | | |
| Initial model used (PDB code) | 6BE1 | 6BE1 | 6BE1 |
| Map sharpening *B*-factor (Å$^2$) | −30 | −30 | −70 |
| Model composition Non-hydrogen atoms Protein residue numbers Ligand atoms | 16,861 393 586 | 16,885 394 585 | 16,885 394 585 |
| *B*-factors (Å$^2$) Protein Ligand | 101.61 103.61 | 115.04 100.96 | 132.89 116.04 |
| R.m.s. deviations Bond lengths (Å) Bond angles (°) | 0.008 0.910 | 0.008 0.991 | 0.009 1.069 |
| Validation MolProbity score Clashscore Poor rotamers (%) | 1.38 (97[th] Percentile) 2.22 (99[th] Percentile) 0.82 | 1.48 (96[th] Percentile) 3.19 (97[th] Percentile) 0.27 | 1.41 (97[th] Percentile) 2.53 (98[th] Percentile) 0.82 |
| Ramachandran plot Favored (%) Allowed (%) Disallowed (%) | 94.40 5.60 0.00 | 94.62 5.38 0.00 | 94.62 5.38 0.00 |

## 5-HT$_{3A}$R model building

The final refined models have clear density of residues Thr7–Leu335 and Leu397–Ser462. The unobserved density at the region of (336–396) is comprised of an unstructured loop which links the amphipathic MX helix and the MA helix. The 5-HT$_{3A}$R-apo cryo-EM structure (PDB ID: 6BE1) was used as an initial model and refined against its EM-derived map using PHENIX software package (*Adams et al., 2002*), using rigid body, local grid, NCS, and gradient minimization parameters. COOT is used for manual model building (*Emsley and Cowtan, 2004*). Real space refinement in PHENIX yielded the final model with a final model to map cross-correlation coefficient of 0.834 (5-HT$_{3A}$R-Palono), 0.846 (5-HT$_{3A}$R-Ondan), and 0.848 (5-HT$_{3A}$R-Alo). Stereochemical properties of the model were validated by Molprobity (*Chen et al., 2010*). The pore profile was calculated using the HOLE program (*Smart et al., 1996*). Figures were prepared using PyMOL v.2.0.4 (Schrödinger, LLC).

## MD simulation setup, protocol, and analysis

The cryo-EM-derived structures of 5-HT$_{3A}$R in the apo conformation or bound to palonosetron, alosetron, ondansetron, or serotonin were prepared for MD simulations with the Protein Prep Wizard in the Schrödinger scientific software suite 2019–2 using default settings (Small-Molecule Drug Discovery Suite 2019–2, Schrödinger, LLC, New York, NY, 2019). This protocol adds missing hydrogen atoms to the initial protein-ligand complex. After the initial preparatory steps and protonation assignment of side chains, a brief restrained energy minimization *in vacuo* using the OPLS3 force field (*Harder et al., 2016*) was carried out to finalize system setup for each protein-ligand complex. Each setron-5-HT$_{3A}$R complex was then embedded into a POPC bilayer using the Membrane Builder tool of the CHARMM-GUI webserver (*Jo et al., 2008*). The system was then solvated with TIP3P water, and 150 mM NaCl was added to the simulation system by replacing random water molecules. Excess sodium ions were added to neutralize the charge of each protein-ligand complex. The resulting simulation systems had initial dimensions of ~130 × 130×207 Å$^3$ and consisted of the unliganded 5-HT$_{3A}$R pentamer, or the pentamer bound to the setron or serotonin at each 5-HT$_{3A}$R subunit,~400 POPC molecules,~83,000 water molecules,~240 sodium ions, and ~220 chloride ions, for a total of ~330,000–346,000 atoms. Throughout this work we reference data from our previously published simulation of granisetron-bound 5-HT$_{3A}$R (*Basak et al., 2019*) in comparison to these three new setron-bound 5-HT$_{3A}$R complexes, as well as the 5-HT$_{3A}$R-serotonin complex and the 5-HT$_{3A}$R-Apo structure.

The CHARMM36m forcefield (*MacKerell et al., 1998*) was used to parameterize the protein and lipid atoms within each simulation system. Initial parameters for palonosetron, alosetron, and ondansetron were obtained from the ParamChem webserver using the CHARMM general force field (*Vanommeslaeghe et al., 2010*) (https://cgenff.parmchem.org). Parameters were validated according to the procedure described previously (*Vanommeslaeghe et al., 2010*). Said validation required quantum calculations performed with Gaussian 16 (Gaussian 16, Revision C.01, Gaussian, Inc, Wallingford CT, 2016) to finalize the charges and dihedrals defined within our setron molecule models. These parameter refinement steps were not conducted for serotonin as the default ParamChem parameters were found to be sufficient as described in *Vanommeslaeghe et al., 2010*.

MD simulations were run using GROMACS 2018.6 (*Berendsen et al., 1995*) software with a time-step of 2 fs, following a steepest descent energy minimization run for 5000 steps, as well as 100 ps isothermal-isovolumetric (NVT) and 52 ns isothermal-isobaric (NPT) equilibration runs. The NVT equilibration was performed to initially heat the model systems after the steepest descents minimization. This step was performed with restraints on protein, membrane, and ligand molecule heavy atoms (when ligand was present) relative to their starting conformation. The NPT equilibration runs were performed in 5 steps of 10 ns each, within which the system was allowed to relax with gradually released restraints until finally the system was allowed to equilibrate for 2 ns of unrestrained NPT equilibration. This was followed by a 100 ns production run in isothermal-isobaric conditions. System temperature and pressure were maintained at 300 K and 1 bar, respectively, using velocity rescale (*Bussi et al., 2007*) for temperature coupling and Parrinello-Rahman barostat for pressure coupling during equilibration. Semi-isotropic pressure coupling and the Nose-Hoover thermostat (*Hoover, 1985*) were applied during production runs. All bonds involving hydrogens were constrained using the LINCS algorithm (*Hess et al., 1997*). Short-range nonbonded interactions were cut at 12

Å. Long-range electrostatic interactions were computed using the Particle Mesh Ewald summation with a Fourier grid spacing of 1.2 Å.

Trajectory analyses were performed using a combination of Visual Molecular Dynamics (VMD) (*Humphrey et al., 1996*) and the GROMACS analysis toolkit (*Van Der Spoel et al., 2005*) over equidistant frames of our production simulations using a 500 ps stride. In particular, all RMSD measurements and Loop C orientations were obtained after aligning simulation frames onto the coordinates of the initial cryo-EM structure by comparing Cα atoms in the helices and β-sheets of the ECD. RMSD calculations were assessed for each ligand by evaluating the difference in heavy atoms of the ligands between each simulation frame and the initial cryo-EM structure conformation. Similarly, Loop C RMSD's were calculated by comparing the Cα, backbone carbonyl carbon, and backbone nitrogen atoms of residues Ser200 through Asn205 relative to their conformation in the initial cryo-EM resolved structures. To measure the orientation of Loop C, we defined a custom Loop C dihedral as being drawn between the alpha carbons of residues Ala208, Phe199, Glu198, and Ile203. To determine whether Loop C adopted a 'closed' or 'open' conformation we evaluated the distance between the Arg65 and Asp202 side chains, measured by a minimum distance of their respective polar side-chain atoms for each analyzed simulation frame. To evaluate how well solvated the setron-binding sites were throughout our simulations, we counted the number of water oxygen atoms within 3 Å of any setron atoms for each simulation frame averaged across all five subunits. Structural interaction fingerprints were calculated with an in-house python script that monitored 5-HT$_{3A}$R interactions with each setron. Specifically, for each residue of 5-HT$_{3A}$R, ligand-protein interactions with both sidechain and backbone heavy atoms were calculated as a 9-bit representation based on the following 9 types of interactions: apolar (van der Waals), face-to-face aromatic, edge-to-face aromatic, hydrogen-bond interactions with the protein either as a donor or acceptor, electrostatic with either the protein acting as a positive or negative charge, one-water-mediated hydrogen bond, and two-water-mediated hydrogen bonds. A distance cutoff of 4.5 Å was used to identify apolar interactions between two non-polar atoms (carbon atoms), while a cutoff of 4 Å was used to evaluate aromatic and electrostatic interactions. Interaction probabilities were averaged across simulation frames as well as across all five 5-HT$_{3A}$R binding sites and errors for each interaction type were estimated using a two-state Markov model, sampling the transition matrix posterior distribution using standard Dirichlet priors for the transition probabilities (*Trendelkamp-Schroer et al., 2015*). Pore radii of 5-HT3AR systems were assessed over equidistant simulation frames with a stride of 12.5 ns using HOLE (*Smart et al., 1996*).

### Data availability accession numbers

The coordinates of the 5-HT$_{3A}$R-setron structures and the corresponding Cryo-EM maps have been deposited in wwPDB and EMDB with the following accession numbers. PDB ID: 6W1Y; EMBD ID: EMD-21518 for 5-HT$_{3A}$R-Palono, PDB ID: 6W1M; EMBD ID: EMD-21512 for 5-HT$_{3A}$R-Ondan and PDB ID: 6W1J; EMBD ID: EMD-21511 for 5-HT$_{3A}$R-Alo.

## Acknowledgements

We acknowledge the use of instruments at the Cryo-Electron Microscopy Core at the CWRU School of Medicine. We are grateful to Dr. Kunpeng Li for assistance with cryo-EM imaging and data collection. We thank Denice Major for assistance with hybridoma and cell culture at the Department of Ophthalmology and Visual Sciences (supported by the National Institutes of Health Core Grant P30EY11373). We thank Dr. Walter F Boron for kindly providing us *Xenopus* oocytes and for unrestricted access of the oocyte rig. We are deeply appreciative of the support provided by Dr. Fraser Moss and Mr. Brian Zeise with the oocyte rig. We are very grateful to the members of the Chakrapani lab for critical reading and comments on the manuscript. Computations were run on resources available through the Office of Research Infrastructure of the National Institutes of Health under award numbers S10OD018522 and S10OD026880, as well as the Extreme Science and Engineering Discovery Environment under MCB080077 (to MF), which is supported by National Science Foundation grant number ACI-1548562. This work was supported by the National Institutes of Health grants R01GM108921, R01GM131216, R35GM134896, and Cryo-EM supplements: 3 R01GM108921-03S1, R01GM108921-5S1, 3 R01GM131216-1S1 to SC and the AHA postdoctoral Fellowship to AK (20POST35210394) and SB (17POST33671152).

## Additional information

### Funding

| Funder | Grant reference number | Author |
|---|---|---|
| National Institute of General Medical Sciences | R01GM108921 | Sudha Chakrapani |
| American Heart Association | 17POST33671152 | Sandip Basak |
| American Heart Association | 20POST35210394 | Arvind Kumar |
| National Institute of General Medical Sciences | R35GM134896 | Sudha Chakrapani |
| National Institute of General Medical Sciences | R01GM131216 | Sudha Chakrapani |

The funders had no role in study design, data collection and interpretation, or the decision to submit the work for publication.

### Author contributions

Sandip Basak, Conceptualization, Data curation, Formal analysis, Funding acquisition, Validation, Investigation, Visualization, Methodology, Writing - original draft, Writing - review and editing; Arvind Kumar, Data curation, Writing - review and editing; Steven Ramsey, Data curation, Formal analysis, Validation, Investigation, Visualization, Methodology, Writing - original draft, Writing - review and editing; Eric Gibbs, Abhijeet Kapoor, Formal analysis, Writing - review and editing; Marta Filizola, Formal analysis, Supervision, Funding acquisition, Validation, Writing - original draft, Writing - review and editing; Sudha Chakrapani, Conceptualization, Formal analysis, Supervision, Funding acquisition, Validation, Investigation, Visualization, Methodology, Writing - original draft, Project administration, Writing - review and editing

### Author ORCIDs

Sandip Basak (iD) https://orcid.org/0000-0003-4018-8020
Arvind Kumar (iD) http://orcid.org/0000-0002-8421-8669
Steven Ramsey (iD) https://orcid.org/0000-0001-7441-3228
Abhijeet Kapoor (iD) https://orcid.org/0000-0002-3606-3463
Sudha Chakrapani (iD) https://orcid.org/0000-0003-0722-2338

### Decision letter and Author response

Decision letter https://doi.org/10.7554/eLife.57870.sa1
Author response https://doi.org/10.7554/eLife.57870.sa2

## Additional files

### Supplementary files

- Transparent reporting form

### Data availability

All data generated or analysed during this study are included in the manuscript and supporting files.

The following datasets were generated:

| Author(s) | Year | Dataset title | Dataset URL | Database and Identifier |
|---|---|---|---|---|
| Basak S, Chakrapani S | 2020 | Cryo-EM structure of 5HT3A receptor in presence of Alosetron | http://www.rcsb.org/structure/6W1J | RCSB Protein Data Bank, 6W1J |
| Basak S, Chakrapani S | 2020 | Cryo-EM structure of 5HT3A receptor in presence of Ondansetron | http://www.rcsb.org/structure/6W1M | RCSB Protein Data Bank, 6W1M |

| Basak S, Chakrapani S | 2020 | Cryo-EM structure of 5HT3A receptor in presence of Palonosetron | http://www.rcsb.org/structure/6W1Y | RCSB Protein Data Bank, 6W1Y |

The following previously published datasets were used:

| Author(s) | Year | Dataset title | Dataset URL | Database and Identifier |
|---|---|---|---|---|
| Basak S, Chakrapani S | 2019 | Cryo-EM structure of 5HT3A receptor in presence of granisetron | https://www.rcsb.org/structure/6NP0 | RCSB Protein Data Bank, 6NP0 |
| Zarkadas E, Zhang H, Cai W, Effantin G, Perot J, Neyton J, Chipot C, Schoehn G, Dehez F, Nury H | 2020 | Mouse serotonin 5HT3 receptor in complex with palonosetron | https://www.rcsb.org/structure/6Y1Z | RCSB Protein Data Bank, 6Y1Z |

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
