## [Decision Letter]

**Acceptance summary:**

Serotonin receptors (5-HT_3_Rs) are pentameric neurotransmitter-gated ion channels that play a crucial role in regulating gut movement. Drugs used to manage the nausea and vomiting associated with radiation and chemotherapies work by binding to these receptors. This study reports 3 new cryo-EM structures of full-length 5-HT_3_Rs in complex with 3 different drugs called setrons, which are used as anti-emetics. The resolutions of the structures are among the highest reported for this family of channels and reveal how these therapeutics bind to 5-HT_3_Rs and antagonize receptor function.

**Decision letter after peer review:**

Thank you for submitting your article "High-resolution Structures of multiple 5-HT_3A_R-setron complexes reveal a novel mechanism of competitive inhibition" for consideration by *eLife*. Your article has been reviewed by two peer reviewers, including Cynthia M Czajkowski as the Reviewing Editor and Reviewer #1, and the evaluation has been overseen by Olga Boudker as the Senior Editor.

The reviewers have discussed the reviews with one another and the Reviewing Editor has drafted this decision to help you prepare a revised submission.

Summary:

Serotonin receptors (5-HT_3_Rs) are cation-selective, pentameric neurotransmitter-gated ion channels that play a crucial role in regulating gut movement. This study reports 3 new high resolution cryo-EM structures of full-length 5HT_3_Rs in complex with different competitive antagonists called setrons, which are used as anti-emetics. Molecular dynamic simulations and functional assays were used to assess ligand-protein interactions and ligand induced motions. The structures add significant new information about antagonist binding to 5HT_3_Rs. By comparing structures (apo, antagonist bound and agonist bound), the authors conclude that antagonist binding stabilizes intermediate conformations along the activation pathway and suggest that this is a novel mechanism.

The strength of the paper is the new cryo-EM structures with resolutions among the highest reported for this family of channels. Unexpected differences in loop C highlight the value of obtaining independent structural data, even for compounds with similar antagonist actions that target the same binding site.

Weaknesses noted are with the interpretations arising from the MD simulations, the limited functional data and the lack of detail in the drug-protein interaction fingerprint analyses. The manuscript would benefit from including additional simulations, deeper analyses of the simulations and clearer interpretations of the 5-HT_3_R-setron interaction fingerprints and LigPlot data. In addition, reorganization and reframing of conclusions to highlight how the data in this study provide new mechanistic insights into how setrons bind at 5-HT_3_Rs and how they inhibit channel function are needed.

Addressing the following essential revisions will strengthen the manuscript.

Essential revisions:

1) Simulations should provide mechanistic insights. In general, the simulations demonstrate that setrons remain in binding pocket and orientation does not change (Figure 5) and that loop C does not move appreciably (supplemental Figure 7). The central model, that setrons inhibit 5-HT_3A_ by locking the receptor-particularly loop C-in an intermediate state, would be substantially strengthened by comparing dynamics in apo and serotonin-bound states. Indeed, in light of the variations among static structures shown in Figures 3/4, it seems strange that Figures 5/6/SI do not include analyses of e.g. loop closure, flexibility, or "dihedral" in apo and serotonin simulations, nor of ligand RMSD or hydration for serotonin simulations. Similarly, how do specific setron interactions reported in Supplemental Figures 5/6 compare to serotonin? The conclusion states they involve "overlapping residues" but extend "further into the complementary subunit," a comparison that does not seem directly documented in this work. At least some of these data would seem readily accessible, as the same authors previously reported simulations of apo and serotonin-bound structures on similar timescales. For example, how does the "open" conformation visited by loop C in setron simulations compare to the apo state?

2) The proposal that setrons stabilize an intermediate conformation also raises striking questions as to the dynamics of inhibited structures beyond the orthosteric site. The authors previously reported fluctuations in pore hydration and diameter in apo versus pre-open structures; how do these compare in the apparently intermediate-diameter pores of setron complexes? Does the apparent constriction of the upper-pore region in the presence of setrons, or the generally shifted profile of the lower pore with alosetron, persist in simulations? If alosetron locks loop C in a 5HT-like conformation but with "varying degrees of structural changes in the inner and outer β-strands that translate to small changes in the pore-lining M2 helices," how are these changes reflected in simulations?

3) The constriction in the ECD appears to be the biggest difference between apo-, antagonist- bound conformations and the serotonin pre-open and open conformations. Identify the residues that contribute to the constriction and discuss more thoroughly on how data supports idea that antagonists stabilize intermediate conformations.

4) The authors do not discuss what information is gained from 5-HT_3_R-setron interaction fingerprints in Supplementary Figures 5 and 6 compared to analyzing and critically examining the cryo-EM structures. Discussion of this data is brief and does not provide enough details. The authors state "Notably, the alosetron fingerprints suggests that this compound forms stronger interactions with Asp202 and Trp156 when compared to palonosetron, ondansetron, and granisetron." It is not clear how these analyses are used to quantitate whether an interaction is stronger than another. The interaction maps do not seem to correspond entirely to structural snapshots in e.g. Figure 5. For example, alosetron is depicted with water-bridged H-bonds to sidechains of N101 and E209, yet these interactions are listed as hydrophobic or nonexistent. Depending on the purpose of the interaction map, it also seems odd to limit lists to sidechain interactions, since several emphasized in Figure 5 are with backbone atoms. What mechanistic or drug-design implications can be derived from these maps?

5) Figure 2: Authors do not thoroughly discuss LigPlot analysis. For example, are there any residues identified that are specific for one setron versus another? Granisetron has more interactions compared to others. A discussion on how data from this paper provides a basis for understanding setrons different affinities should be added.

6) The effort to test predictions from simulations using electrophysiology is encouraging, though less than conclusive. Specifically, dynamics are said to implicate differential interactions with R65, at least for alosetron versus other agents; yet, the functional effect of R65A on alosetron appears to be intermediate. It also seems surprising that R65A decreased palonosetron inhibition as much or more than other setrons, when the equivalent mutation was reported to affect granisetron but not palonosetron (in human receptors in human cells). The R65A data only demonstrate that mutation decreases the functional effects of serotonin and all the antagonists tested. The data do not show any differential effects dependent on ligand. The data provide limited new insight into antagonist actions.

7) A more thorough discussion and summary of the data from this study and from previously published work that support the conclusion that antagonists stabilize intermediate conformations along the activation pathway should be added.

[Editors' note: further revisions were suggested prior to acceptance, as described below.]

Thank you for resubmitting your work entitled "High-resolution Structures of multiple 5-HT_3A_R-setron complexes reveal a novel mechanism of competitive inhibition" for further consideration by *eLife*. Your revised article has been evaluated by Olga Boudker as Senior Editor and a Reviewing Editor.

The manuscript has been improved but there are some remaining issues that need to be addressed before acceptance, as outlined below:

1) Please discuss limitations of short 100msec simulations, which may not capture ligand bound structures found under physiological conditions.

2) Please state how many times the simulations were run for each condition. Were replicates run? Were similar results observed?

3) Figure 7 supplemental, MD simulation data indicate that pore radius is not different between any of the conditions. However, structural data shown in Figure 7B indicate pore radius at 9' position is different between apo versus serotonin – state1, setron-bound versus serotonin – open state. Please clarify, the MD simulations are starting from the structures.

4) Subsection “Conformational differences along the ion permeation pathway” paragraph two. In recent higher resolution structure of muscle-type nAChR from Hibbs group (Nature 2020), lateral portals are open in the apo-state. Please add and reference.

5) "Serotonin binds […] elicits Loop C closure, which may be coupled to channel opening." Data in this paper indicate that antagonist binding is also associated with Loop C closure. Please revise sentence. The authors provide no data in this paper to support the conclusion that Loop C closure is coupled to channel opening.

---

## [Author Response]

Essential revisions:1) Simulations should provide mechanistic insights. In general, the simulations demonstrate that setrons remain in binding pocket and orientation does not change (Figure 5) and that loop C does not move appreciably (supplemental Figure 7). The central model, that setrons inhibit 5-HT_3A_ by locking the receptor-particularly loop C-in an intermediate state, would be substantially strengthened by comparing dynamics in apo and serotonin-bound states. Indeed, in light of the variations among static structures shown in Figures 3/4, it seems strange that Figures 5/6/SI do not include analyses of e.g. loop closure, flexibility, or "dihedral" in apo and serotonin simulations, nor of ligand RMSD or hydration for serotonin simulations. Similarly, how do specific setron interactions reported in Supplemental Figures 5/6 compare to serotonin? The conclusion states they involve "overlapping residues" but extend "further into the complementary subunit," a comparison that does not seem directly documented in this work. At least some of these data would seem readily accessible, as the same authors previously reported simulations of apo and serotonin-bound structures on similar timescales. For example, how does the "open" conformation visited by loop C in setron simulations compare to the apo state?

We agree with the reviewers on the importance of comparing the dynamics of the various 5-HT_3A_R-setron complexes presented in this manuscript with that of 5-HT_3A_R-Apo and 5-HT_3A_R-Serotonin. To this end, we carried out new simulations of apo and serotonin-bound conformations (state 1) using the same protocol used for simulations of the 5-HT_3A_R-setron complexes. We report a new plot of the serotonin RMSD (Figure 3A), which suggests that this ligand is more mobile in the binding pocket as compared to setrons. The plots of loop C RMSD and dihedral angles, water interactions, and R65-D202 interactions for 5-HT_3A_R-Apo and 5-HT_3A_R-Serotonin are also included (Figure 4—figure supplement 1; Figure 6A). Taken together, the results of these analyses point to a more stable “closed” conformation of Loop C in all the ligand-bound 5-HT_3A_R structures, including the 5-HT_3A_R-serotonin structure, compared to a much more flexible conformation of this loop in the absence of a ligand. Such enhanced flexibility in the unliganded state has also been reported in a recent 20 μs simulation study.

2) The proposal that setrons stabilize an intermediate conformation also raises striking questions as to the dynamics of inhibited structures beyond the orthosteric site. The authors previously reported fluctuations in pore hydration and diameter in apo versus pre-open structures; how do these compare in the apparently intermediate-diameter pores of setron complexes? Does the apparent constriction of the upper-pore region in the presence of setrons, or the generally shifted profile of the lower pore with alosetron, persist in simulations? If alosetron locks loop C in a 5HT-like conformation but with "varying degrees of structural changes in the inner and outer β-strands that translate to small changes in the pore-lining M2 helices," how are these changes reflected in simulations?

We have investigated the dynamics of the pore within the 5-HT_3A_R-setron structures, as well as the 5-HT_3A_R-serotonin structure, during 100 ns unrestrained simulations and compared it to results of simulations of the 5-HT_3A_R-Apo carried out in the same environment (POPC lipid bilayer in 150 mM NaCl) and using the same protocol (Figure 7—figure supplement 1). No positional restraints were placed on the molecules during the simulation runs and, as expected, there were no major changes to the overall pore profile during the limited simulation timescale. In each case, the pore remained constricted at the Leu260 (9′) to under 2.3 Å (below the hydrated Na^+^ radius). As also reported in earlier simulation studies, Leu260 (9′) position is the major barrier in the transmembrane domain (TMD) to ion permeation in the 5-HT_3A_R-Apo and 5-HT_3A_R-Serotonin structures. These analyses suggest that although there are different extents of TMD conformational changes in the 5-HT_3A_R-setron structures, each of these conformations appear to be non-conducting to ions. We note, however, that no dynamic transitions between multiple states or allostery between ligand-binding at the ECD and pore-opening at the TMD were expected during the limited standard MD timescales. The purpose of our simulations was to verify ligand stability in the pocket rather than ligand-induced conformational rearrangements. The latter would most likely require enhanced MD simulations to observe transitions between different conformational states.

3) The constriction in the ECD appears to be the biggest difference between apo-, antagonist- bound conformations and the serotonin pre-open and open conformations. Identify the residues that contribute to the constriction and discuss more thoroughly on how data supports idea that antagonists stabilize intermediate conformations.

The reviewers make a good point on the ECD constriction. We have expanded on this in the discussion of the results. Briefly, this constriction point is lined by Asp105 and Lys108 in the β4-β5 loop. The Asp105 position is conserved among most cation-selective pLGICs and mutations at this position affect single-channel conductance and open probability in pLGICs. The ECD constriction is narrower in 5-HT_3A_R-Apo and widens in the serotonin-bound structures. 5-HT_3A_R-setron structures show different extents of widening at this position. However, previous studies assessing permeation of ion and water molecules with imposed membrane potential have shown that this constriction point does not impede ion permeation in the 5-HT_3A_R-Apo and 5-HT_3A_R-Serotonin systems.

4) The authors do not discuss what information is gained from 5-HT_3_R-setron interaction fingerprints in Supplementary Figures 5 and 6 compared to analyzing and critically examining the cryo-EM structures. Discussion of this data is brief and does not provide enough details. The authors state "Notably, the alosetron fingerprints suggests that this compound forms stronger interactions with Asp202 and Trp156 when compared to palonosetron, ondansetron, and granisetron." It is not clear how these analyses are used to quantitate whether an interaction is stronger than another. The interaction maps do not seem to correspond entirely to structural snapshots in e.g. Figure 5. For example, alosetron is depicted with water-bridged H-bonds to sidechains of N101 and E209, yet these interactions are listed as hydrophobic or nonexistent. Depending on the purpose of the interaction map, it also seems odd to limit lists to sidechain interactions, since several emphasized in Figure 5 are with backbone atoms. What mechanistic or drug-design implications can be derived from these maps?

The discussion of the drug-receptor section has been reorganized, and it now includes interaction fingerprint analysis results extended to the protein backbone from 100 ns MD simulations of the 5-HT_3A_R-setron structure (see Figure 3—figure supplement 2). The discussion includes details of these interactions with setrons, as well as those with serotonin. We have also highlighted in the text that a number of interactions seen in simulations were not directly evident from the cryo-EM structures given that they were mediated by water molecules. In addition, MD simulations capture several transient interactions arising from side-chain flexibility and drug mobility within the pocket.

5) Figure 2: Authors do not thoroughly discuss LigPlot analysis. For example, are there any residues identified that are specific for one setron versus another? Granisetron has more interactions compared to others. A discussion on how data from this paper provides a basis for understanding setrons different affinities should be added.

We have included an additional LigPlot analysis figure (Figure 2—figure supplement 1) that shows surface representation with color-coding for apolar and polar interactions. As such most of the interactions of binding-site residues with setrons appear to be hydrophobic. We have extended the section on interaction fingerprint from MD analysis that includes water molecules within the binding-pocket. As can be seen from this analysis, many of the interactions are mediated through water molecules that are not evident in the cryo-EM structures.

6) The effort to test predictions from simulations using electrophysiology is encouraging, though less than conclusive. Specifically, dynamics are said to implicate differential interactions with R65, at least for alosetron versus other agents; yet, the functional effect of R65A on alosetron appears to be intermediate. It also seems surprising that R65A decreased palonosetron inhibition as much or more than other setrons, when the equivalent mutation was reported to affect granisetron but not palonosetron (in human receptors in human cells). The R65A data only demonstrate that mutation decreases the functional effects of serotonin and all the antagonists tested. The data do not show any differential effects dependent on ligand. The data provide limited new insight into antagonist actions.

We fully agree with the reviewers that R65A mutational analysis while revealing a functional effect on serotonin and setron binding does not provide conclusive evidence for the differential effects of various drugs. We think that the effect of R65 is exerted in concert with neighboring residues. Additional mutagenesis and combination of mutations may be needed to understand the mechanism better. We have noted this limitation in the text.

7) A more thorough discussion and summary of the data from this study and from previously published work that support the conclusion that antagonists stabilize intermediate conformations along the activation pathway should be added.

Now included in the discussion of AChBP and other pLGIC.

[Editors' note: further revisions were suggested prior to acceptance, as described below.]

The manuscript has been improved but there are some remaining issues that need to be addressed before acceptance, as outlined below:1) Please discuss limitations of short 100msec simulations, which may not capture ligand bound structures found under physiological conditions.

We have included the following text (was from previous response to reviewer comments) in the main manuscript:

“A limitation of these standard, shorter MD simulation timescales is that dynamic transitions between multiple states or allostery between ligand-binding at the ECD and pore-opening at the TMD are not expected to be captured during runs. The purpose of these simulations were to verify ligand stability in the pocket and the overall stability of the cryo-EM conformation rather than ligand-induced conformational rearrangements. Future studies with enhanced MD simulations may provide insights into transitions between different conformational states and the mechanistic details of coupling across domains.”

2) Please state how many times the simulations were run for each condition. Were replicates run? Were similar results observed?

The following details are included in the text:

“To assess the stability of each ligand binding-pose modeled from cryo-EM density, we quantified the root mean square deviation (RMSD) of each pose relative to its starting conformation, assessed for each subunit independently every 500 ps of each ligand-5HT_3A_R 100 ns simulation for a total of 200 simulation frames (Figure 3A). We also quantified the average RMSD of each pose relative to its starting conformation by averaging over 1000 simulation snapshots (200 frames sampled every 500 ps for each of the 5 subunits treated as replicates) extracted from the 100 ns simulations for each ligand-5HT_3A_R complex.”

3) Figure 7 supplemental, MD simulation data indicate that pore radius is not different between any of the conditions. However, structural data shown in Figure 7B indicate pore radius at 9' position is different between apo versus serotonin – state1, setron-bound versus serotonin – open state. Please clarify, the MD simulations are starting from the structures.

The MD simulations are starting from the cryo-EM structures of the Apo, serotonin-state 1, and the respective setron complexes. While there are fluctuations in the pore radii, the pore remains constricted at the 9’ position in simulations of all the structures. Please note that the serotonin-open conformation was not included in the simulation runs. The pore profile of this conformation was added to Figure 7B upon request from the reviewers.

4) Subsection “Conformational differences along the ion permeation pathway” paragraph two. In recent higher resolution structure of muscle-type nAChR from Hibbs group (Nature 2020), lateral portals are open in the apo-state. Please add and reference.

“The extent of occlusion is similar to that seen in the 5-HT_3A_R-Apo structure^28^ suggesting that the ICD exits are closed in these conformations. Interestingly, in muscle-type nAChR, the ICD portals appear to be open even in the Apo conformation, highlighting the mechanistic differences among pLGIC members^55^.”

5) "Serotonin binds […] elicits Loop C closure, which may be coupled to channel opening." Data in this paper indicate that antagonist binding is also associated with Loop C closure. Please revise sentence. The authors provide no data in this paper to support the conclusion that Loop C closure is coupled to channel opening.

“Serotonin binds within a partially solvent-exposed cavity at the subunit interface and elicits Loop C closure and twisting of the β-strands within the ECD.”